# HOMEOSTATIC UBIQUITY OF HEBBIAN DYNAMICS IN REGULARIZED LEARNING RULES

## ABSTRACT

Hebbian and anti-Hebbian plasticity are widely observed in the biological brain, yet their theoretical understanding remains limited. In this work, we find that when a learning method is regularized with L2 weight decay, its learning signal will gradually align with the direction of the Hebbian learning signal as it approaches stationarity. This Hebbian-like behavior is not unique to SGD: almost any learning rule, including random ones, can exhibit the same signature long before learning has ceased. We also provide a theoretical explanation for anti-Hebbian plasticity in regression tasks, demonstrating how it can arise naturally from gradient or input noise, and offering a potential reason for the observed anti-Hebbian effects in the brain. Certainly, our proposed mechanisms do not rule out any conventionally established forms of Hebbian plasticity and could coexist with them extensively in the brain. A key insight for neurophysiology is the need to develop ways to experimentally distinguish these two types of Hebbian observations.

## 1 INTRODUCTION

Hebbian and anti-Hebbian plasticity are the most commonly observed types of plasticity in the brain (Koch et al., 2013; Zenke & Gerstner, 2017; Lisman, 1989; Lamsa et al., 2007). It is a longstanding belief in neuroscience that Hebbian learning is fundamentally distinct from gradient descent (Hebb, 2005). While Hebbian learning is a simple, unsupervised learning rule that is biologically plausible, gradient-based optimization is widely regarded as requiring access to global error signals and precise coordination across layers—properties not generally supported by neural circuits in the brain. As a result, gradient descent has been largely dismissed as biologically implausible (Rumelhart et al., 1986; Whittington & Bogacz, 2019; Lillicrap et al., 2020), despite its centrality to modern machine learning. Although some recent work has hypothesized algorithms that empirically approximate SGD, which could be implemented through local learning rules in the brain (Lillicrap et al., 2020; Liao et al., 2024), there is limited evidence that our brains are actually learning in any of the proposed mechanisms. At the same time, the mechanistic understanding from the neuroscience side on Hebbian learning and anti-Hebbian learning is poor and often phenomenology-driven. For example, in the spike-timing-dependent-plasticity (STDP) theory (Caporale & Dan, 2008; Froemke et al., 2005; Brzosko et al., 2019), the division between Hebbian and anti-Hebbian learning only depends on the timing of firing, and there is a lack of understanding of why this is the case. However, this separation between artificial and biological learning may be less stark than previously thought. There is some apparent resemblance between Hebbian learning and SGD. Hebbian learning requires weight decay or forms of normalization to ensure convergence, as the core Hebbian principle functions primarily as a learning signal (Oja, 1982b). Similarly, in SGD with weight decay, SGD serves as the learning signal, while weight decay acts as a regularization mechanism that promotes robustness and generalization.

In this work, we discover deeper connections between SGD and Hebbian learning. We demonstrate that the standard training routines used in deep learning, especially stochastic gradient descent (SGD) with weight decay and noise, can give rise to learning signals that look Hebbian or anti-Hebbian. Our results demonstrate that:

1. Close to stationarity, almost any learning rule (including SGD) with weight decay will have a learning signal that looks like a Hebbian rule; and the correlation increases monotonically as we use a larger weight decay;

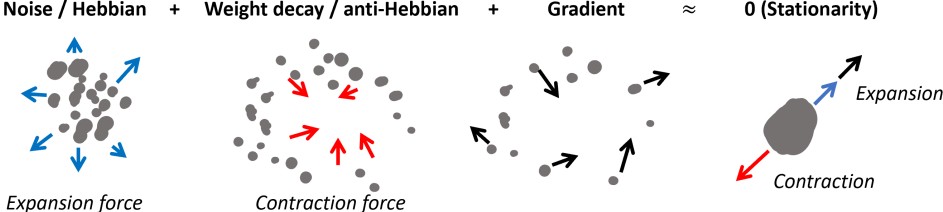

Figure 1: **Balance of contractive and expansive forces.** For deep learning, the noise and weight decay are, respectively, expansion and contraction forces. When they do not balance, the gradient must fill in the gap – if noise outweighs weight decay, the gradient must appear contractive; otherwise, it appears expansive. Similarly, for biology, the Hebbian dynamics is always expansive, and the anti-Hebbian dynamics is always contractive. Thus, to reach a balance, a learning signal will look like, and become aligned with, the Hebbian or anti-Hebbian rule depending on whether it is expansive or contractive.

   2. When we inject noise into the learning process, the learning signal aligns anti-Hebbian, and the effect also becomes stronger as the noise gets stronger;

Although the learning dynamics induced by regularization can resemble those of a mechanistic Hebbian process, they need not involve any Hebbian computation. In other words, there are two distinct paths to dynamics that look Hebbian. Our goal is not to decide which is "correct," nor to speculate about which the brain employs; both may well operate simultaneously. Instead, we present an emergent mechanism based on minimal assumptions and show that it can yield Hebbian-like updates by a different route. The coexistence of these mechanisms in biological systems would complicate attempts to tell them apart experimentally and calls for caution when interpreting synaptic measurements. Their differences deserve closer study.

This work is organized as follows. The next section discusses the closely related works and preliminary concepts to understand our result. Section 3 studies when SGD is similar to the Hebbian rule. Section 4 studies when SGD is similar to the anti-Hebbian rule. Section 5 studies the dynamical and nonstationary aspects of these phenomena. Additional figures are presented in the Appendix.

## 2 RELATED WORKS AND BACKGROUND

**Hebbian Learning.** As a mathematical model, consider a hidden layer in an arbitrary network:

$$h_b = Wh_a(x), \tag{1}$$

where $h_a$ is the postactivation of the previous layer, and $h_b$ is the preactivation of the current layer. In the most conventional form, the simplest homosynaptic update[1] rule states that $W$ is learned according to

$$\Delta W = s\eta h_b h_a^\top, \tag{2}$$

where $s \in \{-1, 1\}$ is the sign of learning. When $s = 1$, the rule is Hebbian, which states that if neuron $i$ causes neuron $j$ to fire, then their connection should be strengthened. Similarly, when $s = -1$, the rule is anti-Hebbian, as it tends to reduce correlation between neurons. $\eta$ is a positive time constant, which we call the "learning rate." In a neuroscience context, Eq. 2 should be regarded a discrete-time approximation to the true underlying continuous-time process, and the rule implies

$$\dot{W} = s\eta h_a h_a^T W, \tag{3}$$

which increases the norm of $W$ when $s > 0$ and decreases it when $s < 0$. Therefore, Hebbian learning in this limit is always expansive, and anti-Hebbian learning is always contractive (see Figure 1 for a visualization). Evidence exists to show that both Hebbian and anti-Hebbian learning exist widely in the brain (Abbott & Nelson, 2000; Magee & Grienberger, 2020). Yet it is not yet clear when the learning is supposed to be Hebbian as opposed to anti-Hebbian. Our theory offers a very simple mechanistic answer to this question.

Classical formulations of Hebbian plasticity show that simple local rules can recover meaningful structure from sensory input, from normalization stabilized PCA in linear models (Oja, 1982a; Sanger, 1989) to higher order feature extraction in nonlinear and BCM style variants (Bienenstock

---

[1]We use this term as a synonym of Hebbian learning.

et al., 1982; Oja, 1991; Cox & Adams, 2009). Biophysically grounded rules such as voltage-based and STDP-inspired plasticity (Clopath et al., 2010) and recurrent networks combining Hebbian excitation with anti-Hebbian inhibition (Zylberberg et al., 2011) demonstrate how decorrelation, whitening, and sparse receptive fields can arise in realistic circuits. More recent unifying work shows that many such rules can implement ICA or sparse coding objectives (Brito & Gerstner, 2016). Related models explain anti-Hebbian learning as a structured mechanism for maintaining excitation-inhibition balance and promoting decorrelation (Vogels et al., 2011; King et al., 2013). Rather than deriving a particular synaptic rule from a chosen unsupervised objective, our analysis asks when generic regularized learning dynamics, including supervised settings, naturally produce update directions that align with Hebbian or anti-Hebbian models near stationarity and thus is compatible with any additional functional account of how Hebbian or Anti-Hebbian learning happens in the brain.

**Gradient Descent in the Brain.** So far, there has not been any strong evidence that the brain could implement and run any form of gradient descent, despite various theoretical proposals (Kolen & Pollack, 1994; Lillicrap et al., 2020; Whittington & Bogacz, 2019; Richards & Kording, 2023)–and observations of Hebbian plasticity are often implicitly regarded as evidence against gradient descent (e.g., see the criticism of heterosynaptic rules in Porr & Wörgötter (2007)). Our theory shows that gradient descent dynamics can lead to dynamics at stationarity that are consistent with the Hebbian phenomenon, and because of this, observations of Hebbian updates are consistent with the existence of more complicated learning rules in the brain.

**Similarity between learning algorithms.** A few works are closely related to ours. Xie & Seung (2003) shows the equivalence of gradient descent to a form of contrastive Hebbian algorithm (CHA). However, CHA is not biologically Hebbian because it is not a homosynaptic rule, required by the Hebbian principle. There have been several other adjacent ideas to modify the Hebbian rule to lead to learning performance similar to gradient descent or even mathematical equivalences to SGD in certain types of models (Scellier et al. (2018); Xiao et al. (2019); Scellier & Bengio (2019); Ernoult et al. (2022)). But these works fail to provide a general relationship between arbitrary models trained with SGD and do not identify the key role of regularization and noise.

More recently, it was shown that heterosynaptic circuits such as feedback alignment or SAL (Liao et al., 2024) can lead to dynamics similar to Hebbian dynamics (Ziyin et al., 2025). However, to the best of our knowledge, no paper has successfully shown any robust equivalence between SGD with weight decay and Hebbian learning. On the machine learning side, a recent theoretical work (Ziyin et al., 2024) suggested that the representation learning in neural networks is governed by the expansion and contraction of representations during SGD training. However, the relevance to Hebbian learning and neuroscience is unclear.

## 3 LEARNING-REGULARIZATION BALANCE PRODUCES HEBBIAN LEARNING

There have been proposals that a balance between Hebbian and anti-Hebbian dynamics must happen for the brain to reach at least some form of homeostasis (stationarity) (Xie & Seung, 2003; Oja, 1982b; Bienenstock et al., 1982). There is a similar effect in gradient-based training in neural networks. The use of weight decay contracts the weights to become smaller, but learning can hardly happen if the weights are too small. Therefore, any model that reaches some level of stationarity in training must have a gradient signal that is expansive and opposed to the contractive effect of weight decay.

For the layer defined in Eq. 1, the full weight update is

$$\Delta W \propto \underbrace{-(\nabla_{h_b(x)}\ell)h_a^T(x)}_{\text{learning signal}} -\gamma W, \tag{4}$$

where $\ell$ is the loss function and $\gamma$ is the strength of weight decay.

Let us first show that close to stationarity, the gradient always looks contractive. Close to a stationary point, the update should be small in expectation: $\mathbb{E}_x[(\nabla_{h_b(x)}\ell)h_a^T(x)] + \gamma W \approx 0$. Thus, after right multiplying the equality by $W^T$ we substitute in Eq. 1 and find that

$$\mathbb{E}_x[(\nabla_{h_b(x)}\ell)h_b^T(x)] = -\gamma WW^\top. \tag{5}$$

**Normalized Weight Update Example**
**Cosine Similarity: 0.930**

**Normalized Weight Update Example**
**Cosine Similarity: 0.051**

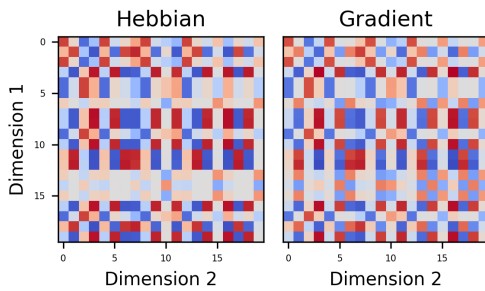

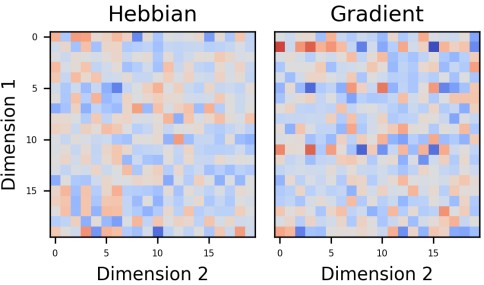

Figure 2: The **left** shows example weight updates with a high alignment between the learning signal $(-\nabla_W \ell)$ and the Hebbian update at the end of training with high weight decay, while the **right** image displays an example update at the end of training with no weight decay which has very low alignment. This figure shows a 20x20 subset of the direction of the Hebbian and learning signal updates for the second layer of an SCE after training with $\eta = 0.1$, and $\gamma = 0.05$, or $\gamma = 0.0$. Dimension 1 can be viewed as the output (post-synaptic) neuron (in the case of SGD, whose incoming weights we are differentiating), and Dimension 2 are input (pre-synaptic) feature/neurons that the weight projects from. We are only visualizing a 20x20 subset of these updates for clarity. Examples of low cosine similarity of the learning signal for $\gamma = 0.05$ at the start and end of training can be seen in Figure 17. In general, we find that stronger weight decay, larger learning rate, and larger batch size lead to better alignment (Figures 7 and 8).

We can then use the Frobenius inner product identity to derive the sign,

$$\mathrm{Tr}\mathbb{E}_x[(\nabla_{h_b(x)}\ell)h_b^T(x)] = \mathbb{E}_x[(\nabla_{h_b(x)}^T\ell)h_b(x)] = -\gamma\mathrm{Tr}[WW^T] < 0. \tag{6}$$

Therefore, on average, $(\nabla_{h_b(x)}^T\ell)h_b(x)$ is negative.

Now, as in Ziyin et al. (2024), we assume a weak decoupling condition: $\|h_a\|^2 = \mathbb{E}[\|h_a\|^2]$, which states that norms of all representations are rather close to each other. This is certainly satisfied when, for example, there is a neural collapse (Papyan et al., 2020) or when the representations are normalized. This means that the expected alignment between the learning signal and the Hebbian update is given by

$$\mathbb{E}\left[\mathrm{Tr}\left[\underbrace{-(\nabla_{h_b(x)}\ell)h_a^T(x)}_{\text{learning signal}}\underbrace{h_a(x)h_b^T(x)}_{\text{Hebbian update}}\right]\right] = -\mathbb{E}[\|h_a\|^2]\mathbb{E}_x[\nabla_{h_b(x)}^T\ell h_b(x)] \tag{7}$$

$$= \gamma\mathbb{E}[\|h_a\|^2]\mathrm{Tr}[WW^T] > 0. \tag{8}$$

Namely, the learning signal has positive correlation with the Hebbian update, and the alignment becomes stronger as $\gamma$ increases. See Figure 2 for an example of such alignment.

Perhaps surprisingly, a weaker form of this result generalizes to any update rule, precisely because the weight decay term always aligns with the anti-Hebbian update; at stationarity the expected learning signal must align with the Hebbian direction. Consider an arbitrary learning signal $g(x, \theta)$; the full weight update is

$$\Delta W = g(x, \theta) - \gamma W, \tag{9}$$

where $g$ is the learning rule and $\theta$ is the entirety of all trainable (plastic) parameters. For clarity, $\eta$ is subsumed into $g$ and $\gamma$. Close to stationarity, we have that $\mathbb{E}_x[g(x, \theta)] \approx \gamma W$.

Now, consider the cosine similarity between the learning rule and the Hebbian rule:

$$\cos\theta = \frac{\langle A, B \rangle_F}{\|A\|_F \|B\|_F} = \frac{\mathrm{Tr}\left[\mathbb{E}_x[g(x, \theta)]\mathbb{E}_x[h_a h_b^T]\right]}{\|\mathbb{E}_x[g(x, \theta)]\|_F \|\mathbb{E}_x[h_a h_b^T]\|_F} \tag{10}$$

The direction of alignment at stationarity when $\mathbb{E}_x[g(x, \theta)] = \gamma W$ is thus

$$\mathrm{Tr}\left[\mathbb{E}_x[g(x, \theta)]\mathbb{E}_x[h_a h_b^T]\right] = \gamma\,\mathrm{Tr}\left[W\mathbb{E}_x[h_a h_b^T]\right] \tag{11}$$

$$= \gamma\mathbb{E}[\|h_b\|^2] > 0. \tag{12}$$

We see that the update must have a positive alignment with the Hebbian rule on average. This shows an intriguing and yet surprising fact: any algorithm with weight decay may look like a Hebbian rule, and the Hebbian rule may just be a "universal" projection of more complicated algorithms. This is a weaker but rather universal result. It is different from Eq. 8 in the sense that Eq. 8 predicts that the learning signal and the Hebbian rule are statistically correlated, whereas this result only says that they have the same direction when averaged over all stimuli. This theory can naturally be extended to the case when the weight decay strength $\gamma$ is different for different neurons, which we discuss in Appendix C.4. Also, this theory can be presented in a fully formal style, which we present in Appendix C.5, where we also formally quantify the time scale and range of the Hebbian dynamics out-of-stationarity.

**Neurobiology.** A key feature of this simple theory is that it separately considers the effects of the learning signal and the weight decay, which, in the context of neurobiology, are likely to have different biological substrates. The learning signal is a fast process; it is likely to, for example, come from other neurons and take the form of electric currents and spikes (Lillicrap et al., 2020). The weight decay, however, is a much slower biochemical process and directly corresponds to the changes in the biochemical properties of synapses (such as a spine shrinkage (Stein et al., 2015)). Therefore, the biological realizations of these two processes are likely to take different forms and can be separately measured. This makes it particularly important to have a theory for the learning-signal component of the update, as this can be directly measured through LTD and LTP experiments of Hebbian plasticity (e.g., see (Zenke & Gerstner, 2017)). We focus on uniform L2 weight decay in this work for its ubiquity in machine learning and analytic simplicity. While Hebbian models often assume some form of L2 weight decay, the brain is unlikely to implement any perfectly uniform weight decay (Oja, 1982a). Although exploring all non-uniform variants of L2 weight decay is too broad for thorough empirical testing, Appendix C.4 extends our analysis to this domain.

**Simulations.** We empirically find that this trend holds across a wide variety of different learning tasks. We ran simulations performing classification on CIFAR-10 and non-linear regression on synthetic data (Krizhevsky, 2009). We tested both MLPs and transformers, as well as a range of activation functions and optimizers. In some situations, the correlation between the two learning paradigms is very strong (e.g., in Figure 2). In our experiments, we used a default learning rate of $\eta = 0.01$ and trained for 50 epochs, which reached convergence. Since this trend only holds near stationarity–a condition achievable in full gradient descent but obscured in SGD by noise–we found it best to use larger batch sizes to compute both the gradient and Hebbian update as suggested in Xu et al. (2023). We found a batch size of 256 to generally show Hebbian phenomena while being small enough to converge to good solutions quickly (Figure 7). To get the alignment between the updates, we compute the cosine similarity of the direction of the gradient update from the loss function (the negative gradient in SGD) and the direction of the Hebbian update. Further, most experiments reported on in this paper followed one of the following setups, and any variations will be reported when relevant.

**(1) Standard Classification Experiment (SCE):** In these experiments, we trained a small MLP with 2 layers of 128 dimensions and tanh activation using cross-entropy loss to classify CIFAR-10.

**(2) Standard Regression Experiment (SRE):** In these experiments, we trained a small MLP with two hidden layers of 128 units each and `tanh` activation, using mean squared error to predict the output of a teacher model. The teacher has the same architecture but is initialized with different random parameters. The input and output vectors are both 32-dimensional, with each element independently drawn from an isotropic Gaussian distribution. The training dataset consisted of 20,000 randomly generated training examples, and the validation dataset contained 2,000 examples. For the transformer variant of the SRE, we used a transformer with 32-dimensional token embeddings, a vocabulary size of 16, and a maximum sequence length of 32. The encoder consists of 2 layers with 4 attention heads and 32-dimensional feed-forward blocks using ReLU activations. The average of the output token embeddings is passed through the same MLP described above and compared to the teacher output.

**Classification.** We train a series of MLPs using the SCE setup to classify CIFAR-10. Figure 3 shows that as weight decay increases, so does the alignment of the learning signal with the Hebbian update. The trend persists across different activation functions. Although we still detect this trend in larger MLPS (Figure 3), we occasionally observe some layers behaving in an anti-Hebbian direction as the weight increases. Using residual connections and batch normalization can stabilize the

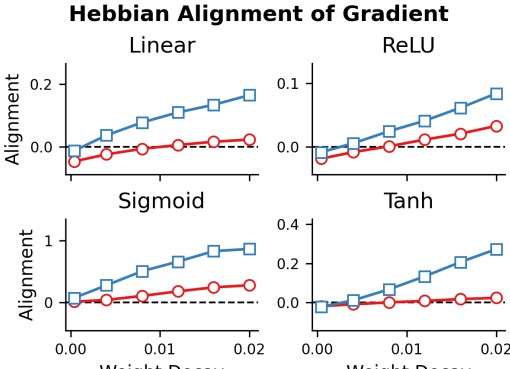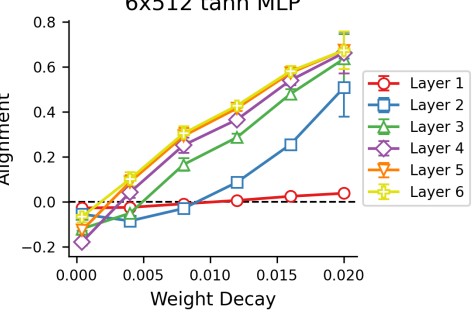

Figure 3: The diagram on the **left** shows that the trend of weight decay increasing Hebbian alignment of the learning signal is robust across different activation functions. The diagram on the **right** shows that the trend can generalize to deeper networks. The SCE MLPs were modified by varying the activation functions across Linear, ReLU, Sigmoid, and Tanh (**left**) and increasing the depth to 6 and layer width to dimension 512 (shown by the 6x512 tanh MLP plot on the **right**). Layer 1, Layer 2, ... Layer 6 in this diagram indicate the Hebbian alignment with the learning signal for the corresponding layer. For a small (or zero) weight decay, the learning process sometimes exhibits a weak anti-Hebbian alignment, indicated by a negative alignment with Hebbian learning. All markers represent the average alignment over the final 100 steps, averaged across 10 runs with different seeds. The error bars represent the std of the final average alignment across the seeds. The trend is very robust, and so many of the bars are obscured by the markers, particularly in the left diagram, for which the largest std was 0.012.

network and make it more Hebbian; however, a deeper exploration of this quality is left for future research. We also explore the use of other regularizations in Appendix 16. By contrast, when we train the same architecture with a Hebbian rule such as Oja's on the same data, the model performs poorly and its learning signal does not align with that of SGD at convergence (Figure 18) (Oja, 1982b). This shows that classical unsupervised PCA-style Hebbian learning does not reproduce the supervised dynamics we study, and that the Hebbian-like signatures we observe are a consequence of regularized supervised optimization rather than explicit PCA feature extraction.

**Regression.** We also evaluate the generalization of this trend to student-teacher regression problems as described in SRE. We explored both MLP and Transformer models and evaluated the Hebbian alignment for learning rules outside of SGD. Recall that a key prediction of the theory is that almost any update look like a Hebbian rule when regularized. We test a variety of rules: SGD, Adam, and Direct Feedback Alignment (DFA) (Nøkland, 2016). To demonstrate that this observation is universal, we also run a setting with a randomly initialized neural network whose output is used as a learning signal, based entirely on the input data (Random NN). Notably, Random NN should not be able to *learn anything* given it is effectively only a deterministic random error signal. Results are shown in Table 1. In all cases, alignment to Hebbian learning emerges and becomes stronger as weight decay increases, regardless of the model.

## 4 Noise-Learning balance leads to Anti-Hebbian learning

We have answered the question of how Hebbian learning can be an emergent and phenomenological byproduct. The second part of the question is when we will see anti-Hebbian learning, as both Hebbian and anti-Hebbian learning are ubiquitous in the brain. Can anti-Hebbian learning also be a byproduct of more complicated learning rules?

The analysis in the previous section does not take into account the existence of noise in learning. In reality, noise is always non-negligible both in biological learning and in artificial learning. That a strong noise leads to an anti-Hebbian learning signal can already be explained by looking at a simple linear regression problem:

$$\ell(w) = (w^T x - y)^2, \tag{13}$$

where $x \in \mathbb{R}^d$, $y \in \mathbb{R}$ are sampled from some underlying distribution at every training step. Here, we inject noise $\epsilon \in \mathcal{N}(0, \sigma I)$ to the weight before every optimization step so that $w = v + \epsilon$, where $v$

Table 1: For all models, optimizers and learning rules, Hebbian alignment rises with increasing weight-decay $\gamma$. Hebbian alignment (mean ± SD, $n$ seeds = 10) at convergence is shown for the 2nd-layer gradient in a regression MLP and a sequence-to-vector transformer (1st layer for DFA). All experiments were SREs with a few modifications outside of the learning rule and weight decay specified in the table. DFA used $\eta = 0.1$ with gradient-norm $clip = 5$ and, as in the original implementation, used biases. RandomNN used gradient-norm $clip = 1$ and a target weight L2 norm of 100 to determine the sign of the update as explained in Section C.6 of the Appendix. Table elements with – indicate the model's weights collapsed to zero.

| Model | Learning Rule | Weight Decay ($\gamma$) | | | |
|---|---|---|---|---|---|
| | | 0 | $5 \times 10^{-5}$ | $5 \times 10^{-4}$ | $5 \times 10^{-3}$ |
| Regression MLP | Adam | $-0.02 \pm 0.00$ | $0.10 \pm 0.00$ | $\mathbf{0.66 \pm 0.01}$ | – |
| | SGD | $-0.10 \pm 0.01$ | $-0.06 \pm 0.01$ | $0.17 \pm 0.01$ | $\mathbf{0.59 \pm 0.01}$ |
| | DFA | $0.45 \pm 0.05$ | $0.45 \pm 0.04$ | $0.68 \pm 0.05$ | $\mathbf{0.87 \pm 0.00}$ |
| | RandomNN | $0.00 \pm 0.00$ | $0.00 \pm 0.00$ | $0.05 \pm 0.00$ | $\mathbf{0.50 \pm 0.00}$ |
| Transformer | Adam | $-0.02 \pm 0.02$ | $0.50 \pm 0.24$ | $\mathbf{0.99 \pm 0.02}$ | – |
| | SGD | $0.00 \pm 0.01$ | $0.04 \pm 0.01$ | $0.47 \pm 0.06$ | $\mathbf{0.88 \pm 0.03}$ |
| | DFA | $0.08 \pm 0.03$ | $0.07 \pm 0.02$ | $0.11 \pm 0.02$ | $\mathbf{0.12 \pm 0.02}$ |
| | RandomNN | $0.00 \pm 0.00$ | $0.00 \pm 0.00$ | $0.01 \pm 0.00$ | $\mathbf{0.09 \pm 0.01}$ |

Figure 4: As the noise increases, the Hebbian alignment decreases and higher weight decays lead to higher Hebbian alignment (**right**). The figure on the **left** displays a heatmap of the Hebbian alignment of the learning signal at convergence for a number of different additive noises and weight decays; there is a clear quadratic curve at zero-alignment as predicted by the theory. The SRE was augmented by adding noise to each parameter at the start of each iteration with a mean of zero and the specified standard deviation on the diagram. The trend is even clearer when we follow the behavior of varying the noise of a specific weight decay (Varying Noise) or the weight decay of a specific noise standard deviation (Varying Weight Decay). Each cell on the left and marker on the right represents a single run.

is the weight before noise injection. This could be a thermal noise that can exist ubiquitously in the brain (London et al., 2010). It can also be seen as an approximate model of the SGD noise, which causes $w$ to fluctuate around the mean (Liu et al., 2021). The learning signal and Hebbian update are

$$\Delta_{\text{SGD}} w = -x(w^T x - y), \tag{14}$$

$$\Delta_{\text{Hebb}} w = x w^T x. \tag{15}$$

The alignment between the two is

$$\mathbb{E}_\epsilon[(\Delta_{\text{SGD}} w)^T (\Delta_{\text{Hebb}} w)] = -\|x\|^2 \mathbb{E}_\epsilon\left[(w^T x)^2 - w^T x y\right] \tag{16}$$

$$= -\|x\|^2 \left[(v^T x)^2 + \sigma^2 \|x\|^2 - v^T x y\right], \tag{17}$$

which is negative for sufficiently large $\sigma^2$ and any $\|x\| \neq 0$. Thus, large noise leads to anti-Hebbian learning.

An interesting question is how this effect competes and trades off with weight decay. When there is a weight decay, the full weight update is $\Delta_{\text{SGD}} w = -x(w^T x - y) - \gamma w$, and so

$$\mathbb{E}_\epsilon[(\Delta_{\text{SGD}} w)^T (\Delta_{\text{Hebb}} w)] \approx -\sigma^2 c_0 + \gamma c_1, \tag{18}$$

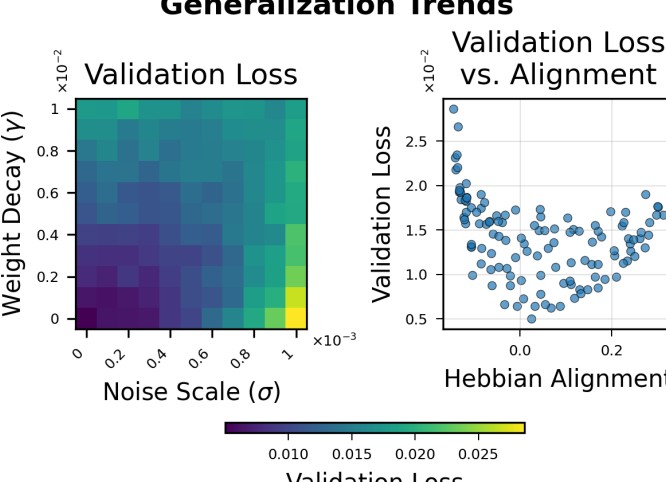

Figure 5: Best performance of the model is achieved when it is not Hebbian or anti-Hebbian on average. The **left** image displays the student validation loss for the experiment in Figure 4, while the **right** image shows a scatter plot of the validation loss vs. Hebbian alignment of the gradient. There seems to be some weak saddle phenomena in loss that occur at the phase transition boundary of Hebbian alignment with respect to noise and scale. The validation loss reduces as both weight decay and noise get smaller. Each cell on the left, and circle on the right, represents a single seed.

where $c_0$ and $c_1$ are positive coefficients that depend on the network architecture and data distribution so can be treated as constants with respect to the weight decay and noise. Thus, one expects a **phase transition** boundary at $\gamma \propto \sigma^2$. When $\gamma$ is larger than this boundary, the learning is Hebbian-like; when smaller, the learning is anti-Hebbian like. This result provides a straightforward and simple framework to potentially test and understand the Hebbian and anti-Hebbian plasticity in biology. A possible strong biological evidence that would verify this theory is the simultaneous observations of strong noise in the ambient space and anti-Hebbian plasticity. In simulation, this scaling law is verified in the experiments (Figure 4), which justifies this simple analysis.

**Simulations** We ran experiments to validate the noise prediction using a two-layer MLP with tanh activation. We used a student-teacher model to build a non-linear regression problem and trained until convergence using SGD and varying the variance of the Gaussian noise added at each training step, as well as the weight decay. There is a very smooth alignment trend with SGD, as can be seen in Figure 4. The white region shows the phase boundary between the Hebbian phase and anti-Hebbian phase, and shows a shape in accordance with the quadratic curve $\gamma \approx \sigma^2$.

We observed that at convergence, the Hebbian alignment of the learning signal is higher in low noise environments, and becomes more aligned with anti-Hebbian as the noise increases (Figure 4). Interestingly, we found that solutions with high generalization generally had low Hebbian and anti-Hebbian alignment (Figure 5).

We also observed this trend with other optimizers such as Adam (Figure 13). However, we struggled to robustly reproduce this effect outside of the last few layers of much larger networks or those doing different learning tasks, such as classification. We hypothesize this could be because the magnitude of the weights does not have as much of an effect on the quality of the learned representations in larger non-linear networks, so the gradient signal does not necessarily need to point in a direction that contracts weights. We also find that adding other types of biologically plausible constraints during learning, such as a sparsification term on layer activations, can lead to a stronger anti-Hebbian alignment of the gradient.

## 5 TRANSIENT PHASES OF HEBBIAN AND ANTI-HEBBIAN LEARNING

As we mentioned in Section 3, the results are applicable when the dynamics are not yet fully stationary. While the argument we had suggested that one would only observe the Hebbian alignment close to convergence, our empirical results suggest that the alignment is present for much of training. Two key phenomena we discover are the initial alignment bump and the steady state Hebbian oscillations.

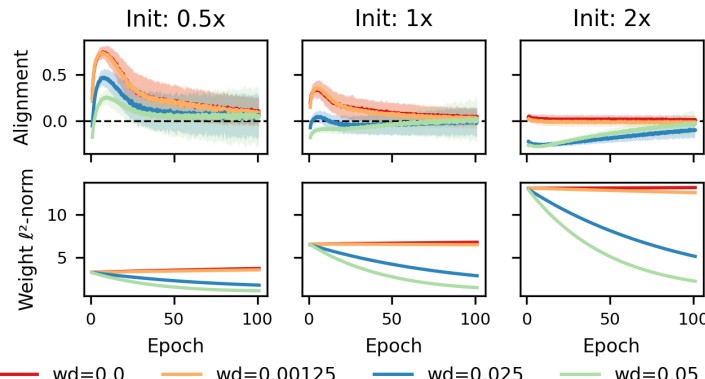

Figure 6: For some activations at low learning rates, there is a sharp jump in Hebbian alignment of the weight update when training with SGD; the size of this jump depends on initial conditions. During this phase, the weight norm decreases monotonically, suggesting the effect is due to feature alignment rather than parameter scale. This experiment used a SCE with $\eta = 0.001$. Init: 0.5x, 1x, and 2x indicate the constant that is used to scale the default torch initialization. The plots above show single seeds to better demonstrate the evolution over time, but the trend is persistent across different seeds.

Outside stationarity, the learning signal often dominates regularization. So it is sufficient to consider the full weight update directly.

**Initial Hebbian alignment bump** Particularly for networks with ReLU activations, there is a bump in Hebbian alignment of the learning signal that appears to be strongly dependent on initialization scale and learning rate at the beginning of training (Figure 6). During this initial phase of alignment, the full weight update of SGD also increases in alignment to a pure Hebbian update. This early stage of alignment seems to be the result of general feature learning, as the actual scale of weight norms does not change substantially at the start of this period, and with positive weight decays decreases. A higher learning rate makes this process happen faster.

When we examine the Hebbian alignment of the weight updates for individual neurons in the model, a striking pattern appears. During this period, individual neurons seem to take on Hebbian or anti-Hebbian learning roles that can persist for many steps (see Figure 20). Like the average behavior of the model, the length of these phases increases as the weight initialization scale magnitude and learning rate decrease. The ratio of anti-Hebbian to Hebbian neurons increases with weight decay.

**Hebbian and Anti-Hebbian steady state oscillations** There is a phase change that occurs after the initial coherent phase of learning, which is accompanied by a strong shift in the magnitude of the alignment intensity. After this phase change, individual neurons often, though not always, seem to oscillate between strongly Hebbian or anti-Hebbian weight updates (Figure 19). Since near stationarity, the magnitude of the parameters should not increase or decrease on average, we also find the mean of the full weight updates to converge to zero (Figure 15). Often, we find that models with better generalization exhibit strong Hebbian/anti-Hebbian oscillations; however, strong oscillations do not necessarily entail strong generalization.

## 6 DISCUSSION

This study suggests that Hebbian and anti-Hebbian plasticity can be understood as emergent regimes of gradient-based optimization rather than fundamentally distinct learning principles. By analyzing the interaction between stochastic gradient descent, weight decay, and stochastic perturbations, we demonstrated that the expected gradient update direction aligns with classic Hebbian plasticity when contraction due to regularization dominates, and switches to an anti-Hebbian alignment when expansion driven by noise prevails. The resulting phase boundary satisfies a simple scaling relation, and the phenomenon was observed across a broad spectrum of architectures, objectives, and alternative update rules.

There are a few limitations that provide interesting areas for future research. We only dealt with smaller models in our experiments. While this decision was reasonable given the scope of this paper, it leaves open the question of whether or not we see Hebbian dynamics in much larger-scale models. As mentioned in the text, as we expanded our models and used different optimizers, we often saw strong average anti-Hebbian alignment of a subset or all of the layers, even at high weight decays. This likely results from our stationarity condition not holding in these models. However, we do not yet have a theory for why and when these regions of anti-Hebbian alignment occur.

Our results have two primary implications. First, our results show that Hebbian and anti-Hebbian plasticity can emerge as regimes of gradient-based optimization, in addition to their conventional role as fundamental learning mechanisms. Second, the presence of Hebbian or anti-Hebbian signatures in neuro-physiological data need not be interpreted as evidence against global error-driven optimization in the brain; such local plasticity patterns may arise as epiphenomena of an underlying optimization process.

Given that unsupervised adaptation and reinforcement are useful and widespread mechanisms, intrinsically Hebbian homosynaptic plasticity likely does exist in the brain. However, much of the existing experimental evidence for Hebbian and anti-Hebbian plasticity is often correlational and phenomenological (e.g., see Lamsa et al. (2007)), so it can be difficult to decide whether the underlying dynamics are actually Hebbian or are more complicated and only appear to be Hebbian. Although some evidence has established the importance of heterosynaptic modulation in memory storage and visual discrimination, its broader role in learning has remained largely speculative due to the challenges of studying it both in vivo and in vitro (Bailey et al., 2000; Chasse et al., 2021). We hope the theory we propose in this paper will serve as a basis for future experimental studies that will validate or challenge the existence of heterosynaptic learning principles in the brain.

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

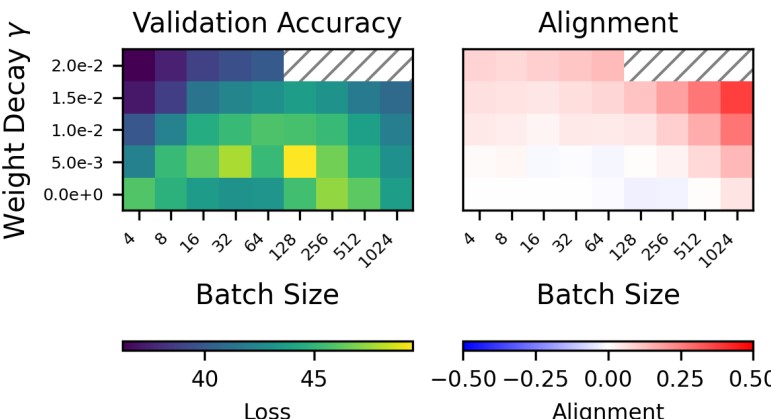

Figure 7: The optimal performance seems to be at a critical position between Hebbian and anti-Hebbian gradient alignment when varying batch size and weight decay. This shows the accuracy (**left**) and the Hebbian alignment of gradient update (**right**) for SCEs with a variety of weight decays and batch sizes. The striped background indicates NaN values.

## A    REPRODUCTION

All experiments were run on MIT's OpenMind cluster using Quadro RTX 6000 GPUs and cumulatively took under 50 hours of compute time.

## B    LLM USAGE

The authors used LLMs to assist in editing the manuscript and writing experimental code.

## C    EXPERIMENTS

In the following document, we provide additional figures and explanations that were referenced in the main text.

### C.1    ADDITIONAL INFLUENCES ON HEBBIAN ALIGNMENT AND GENERALIZATION

#### C.1.1    BATCH SIZE

See Figure 7.

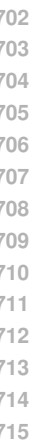
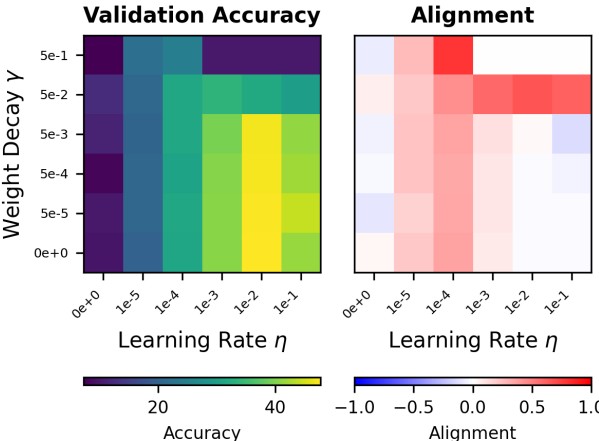

Figure 8: The optimal performance seems to be at a critical position between Hebbian and anti-Hebbian gradient alignment when varying learning weight and weight decay. This shows the accuracy (**left**) and the Hebbian alignment of gradient update (**right**) for SCEs with a variety of weight decays and learning rates.

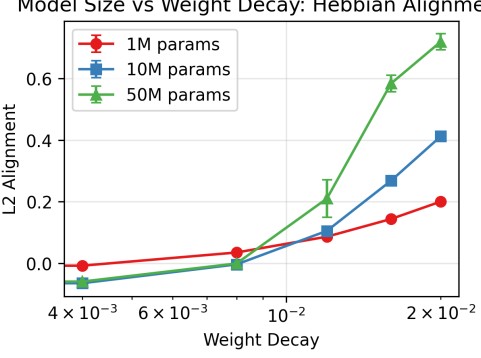

Figure 9: This diagram shows the effect of model size on Hebbian alignment and weight decay. Each point represents the mean of the alignment for the final 200 steps of the run ± the std across 10 seeds. The trend of weight decay leading to increased Hebbian alignment of the learning signal holds with larger models as well. The diagram above shows the alignment of the second layer of the respective MLPs. The MLPs had the following number of total layers: 3, 7, and 9 for the 1M, 10M, and 50M models, respectively. All hidden dimensions were assigned to reach the target parameter count as closely as possible. See equation 19 for how the exact hidden dimensions were computed. The trend is very robust so a number of the error bars are obscured by the markers.

### C.1.2 LEARNING RATE

See Figure 8.

### C.1.3 MODEL SCALE

See Figure 9.

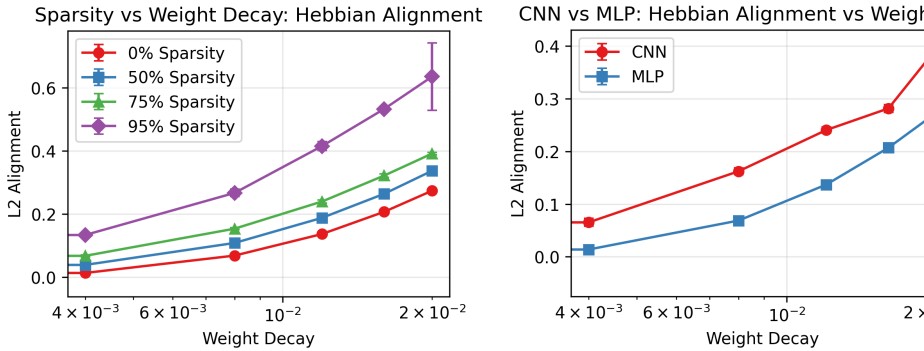

Figure 10: Empirically, the Hebbian alignment of the learning signal increases with sparsity **(left)**. We also see that the linear layers in a convolutional neural network, which are highly sparse, have increased Hebbian alignment. Each point represents the mean of the alignment of the second MLP layer for the final 200 steps of the run ± the std across 10 seeds. The models on the left were the standard MLPs with varying sparsity. The MLP on the right was a standard MLP and the CNN had the following architecture: Conv Layers: $(c_{in} = 3, c_{out} = 32, s = 3, p = 1)$, $(32, 64, 3, 1)$, $(64, 128, 3, 1)$ MaxPool: $(2, 2)$ Linear hidden dimensions: 2048, 512, 256. The trend is very robust so a number of the error bars are obscured by the markers.

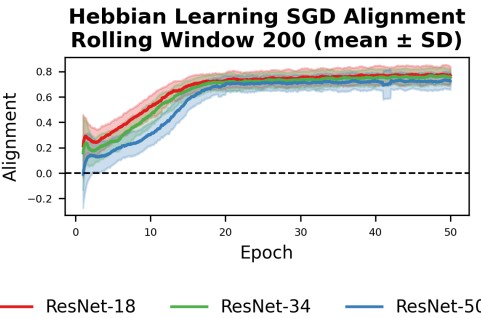

Figure 11: This figure shows an example SCE run with an identical training and model setup as the convolutional network described in Figure 10 but using different ResNet models as the backbone instead.

$$
h = \begin{cases}
\dfrac{-(i+o) + \sqrt{(i+o)^2 + 4t}}{2}, & t = 10^6 \\[2ex]
\dfrac{-(i+o) + \sqrt{(i+o)^2 + 20t}}{10}, & t = 10^7 \\[2ex]
\dfrac{-(i+o) + \sqrt{(i+o)^2 + 28t}}{14}, & t = 5 \cdot 10^7
\end{cases}
\tag{19}
$$

Where $i$ is the dimension of the input, $o$ is the dimension of the output and $t$ is the target parameter count.

### C.1.4 MODEL SPARSITY

See Figures 10 and 11.

### C.1.5 FROZEN PARAMATERS

See Figure 12

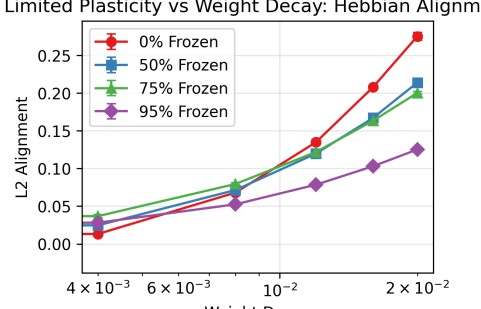

Figure 12: The alignment decreases as the fraction of parameters of the standard MLP that are frozen increases; however, the trend still persists. Each point represents the mean of the alignment for the final 200 steps of the run ± the std across 10 seeds. The trend is very robust so a number of the error bars are obscured by the markers.

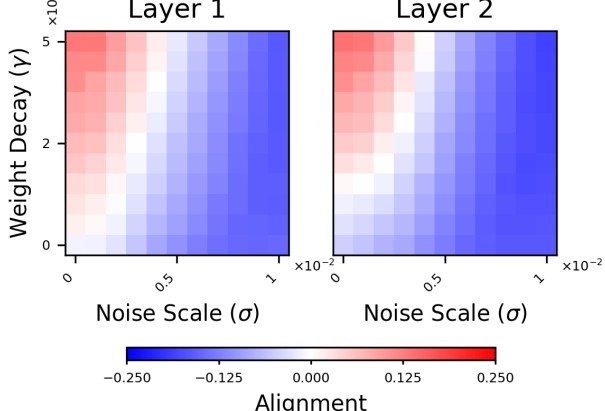

Figure 13: Again, there is a clear trend that even for the Adam optimizer, as noise increases, alignment of the learning signal decreases, and as weight decay increases, so too does alignment. Adam was very sensitive to the parameter ranges for which we'd see the trend, so we used a different weight decay and standard deviation range than the prior experiment. However, the rest of the architecture and experimental setup are identical to that described in Figure 4.

### C.1.6 TRAINING DURATION

### C.1.7 NOISE

See Figure 13 and 14.

### C.1.8 FULL UPDATE VS. LEARNING SIGNAL

As defined in the terminology section, the learning signal $g(x, \theta) \equiv -\nabla_W \ell(\cdot)$ represents the gradient contribution to weight updates, while the full weight update $\Delta W = \eta(g(x, \theta) - \gamma W)$ includes both the learning signal and regularization terms. While we see that the learning signal aligns on average with the Hebbian update, the full weight update can not, otherwise the weight would explode. See Figure 15 for a visualization of the alignment of the learning signal and the full weight update over the course of training. Still, we see a very interesting trend where the full weight update often has strongly Hebbian or anti-Hebbian updates that, on average, cancel.

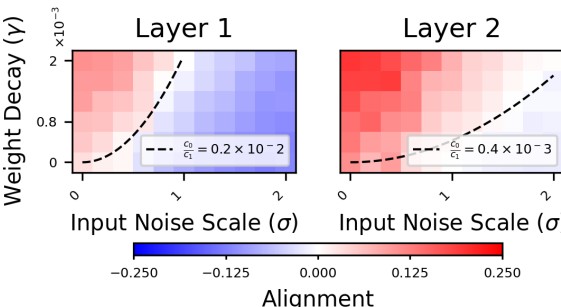

Figure 14: Additive noise to the input can also lead to anti-Hebbian learning. Since noise is only added to the input of the network, the exact phase boundary changes with depth. The results depicted above are from a SRE with input noise injected.

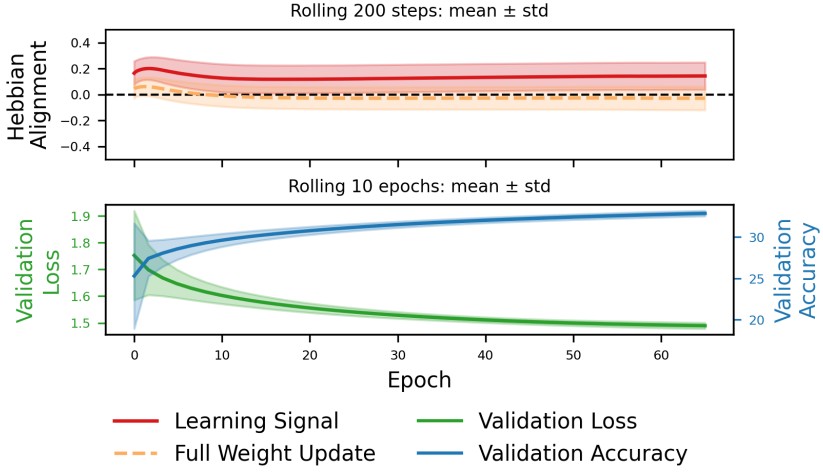

Figure 15: Comparison of Hebbian alignment for learning signal vs. full weight update during training (**top**) and the corresponding validation loss and accuracy (**bottom**). The learning signal alignment shows the characteristic patterns described in the main text, while the full weight update alignment approaches zero near stationarity as expected, since the mean of the full update must be zero at convergence. Individual neuron-wise signals can still oscillate between Hebbian and anti-Hebbian phases even when the mean full update is zero. Learning Signal alignment begins and persists far before learning has stopped. The Hebbian alignment is relatively low given the the low base weight decay used for SCE but reaches a non-trivial positive alignment long before convergence and while useful features are still being learned. The above graph is sampled from a single run to show the evolution over time of the alignment, though this trend is very consistent.

### C.1.9 OTHER REGULARIZATION TECHNIQUES

See Figure 16.

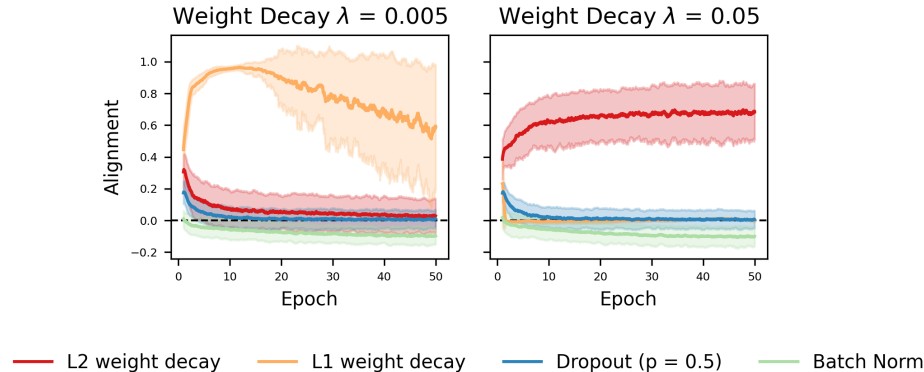

Figure 16: Other regularization techniques have a variety of effects on the Hebbian alignment of the learning signal. While we only developed a theory for L2 weight decay, the alignment seems to exist for some other regularizers as well when used to augment SCEs. Batch normalization seems to have a modest but persistent anti-Hebbian effect, while both L1 and L2 weight decay can have a Hebbian effect, and Dropout has no effect. While the trends above do seem robust across other seeds, the plots above show the evolution of single seeds over time to better visualize the evolution of the alignment throughout training. We present these qualitative findings for completeness; a deeper analysis of other regularization techniques is outside the scope of this work.

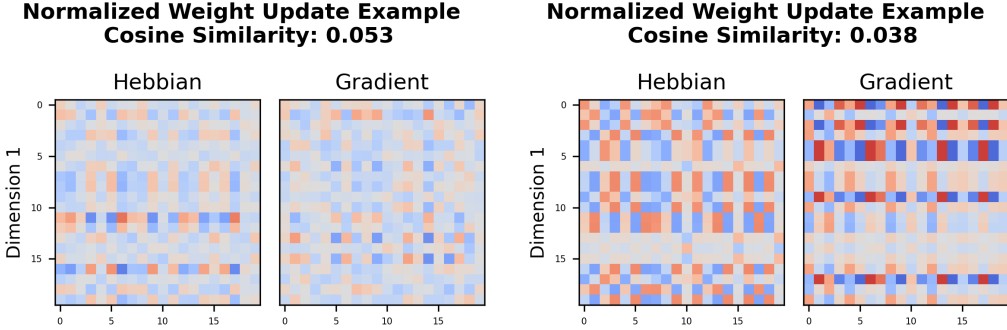

Figure 17: With weight decay, even after the first epoch (**left**), there starts to be an alignment of the directions; at convergence (**right**), even when specific steps have low cosine similarity, there is still clearly a lot of similar structure. At the end of training, many learning signals with low Hebbian alignment still share a surprising amount of structure. The plots above are from a SCE with $\eta = 0.1$ and $\gamma = 0.05$.

## C.2 EXAMPLE ALIGNMENTS DURING TRAINING

### C.2.1 LOW ALIGNMENT UPDATE AT END OF TRAINING

See Figure 17.

## C.3 HEBBIAN LEARNING DOES NOT LEAD TO GRADIENT ALIGNMENT

See Figure 18.

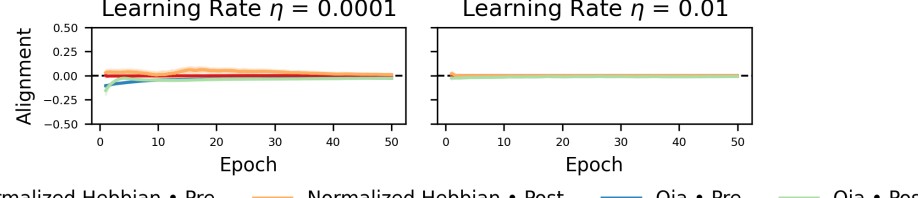

Figure 18: No standard interpretation of Hebbian learning produces alignment with SGD at convergence. The plots above show a different learning setup than the standard SCE; rather than training with SGD and computing the alignment of the learning signal with a Hebbian update at each update step, instead the model is trained with various common interpretations of the Hebbian learning objective then this update is compared to the supervised loss gradient at each step computed with back propagation. The graph shows the mean SGD alignment of the second layer's updates, ± the standard deviation over a 200-iteration window, when trained with various versions of the Hebbian learning rule for two different learning rates. While we found the above trends to hold robustly across various seeds, each line represents only a single run smoothed over time to better demonstrate the evolution of the learning rules with respect to time. The *Normalized Hebbian* learning rule is the generic Hebbian algorithm with weight standardization after every step. The second algorithm is Oja's rule. We also tested the pre-activation and post-activation versions of both. The average alignment of every combination approaches zero.

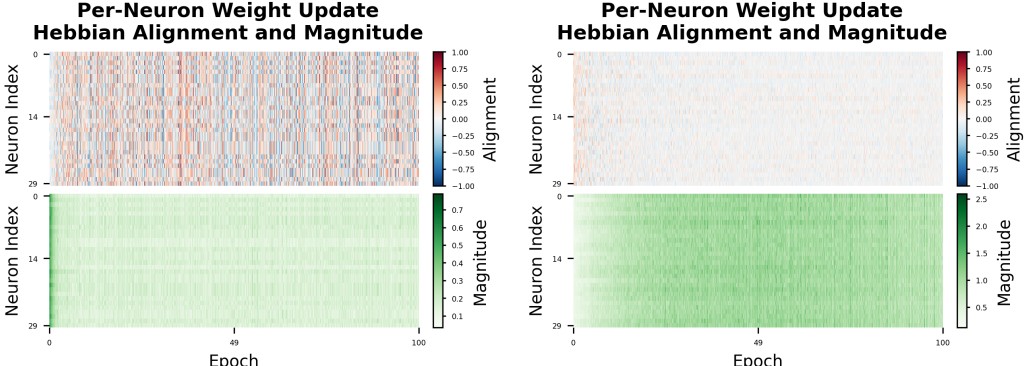

Figure 19: The full weight update of neurons in the neural network strongly oscillates between positively and negatively aligned to the Hebbian update late into training with weight decay (**left**). As can be seen by the blue and red stripes, there is some form of global coherent oscillation in alignment at higher weight decays. This phenomenon becomes much weaker without weight decay (**right**). Both diagrams show 30 example neurons from the second hidden layer of an MLP trained on the standard regression experiment over the course of training. The diagram on the left has a $\gamma = 0.05$ while the one on the right has a gamma of $0.0$, both used tanh activations. Our experiments suggest that the oscillation in alignment for individual neurons is not related to the magnitude of the weight update that neuron is receiving, though the experiment run without weight decay does have higher magnitude weight updates since the weights are larger.

### C.3.1 FULL WEIGHT UPDATE HEBBIAN OSCILLATION

See Figures 19 and 20.

### C.4 NON-UNIFORM L2 WEIGHT DECAY

In the main text, we discussed for notational simplicity the case when the weight decay is uniform across all neurons. The argument can be similarly and simply extended to the case where different neurons have different rates of weight decay. We tackle that situation here.

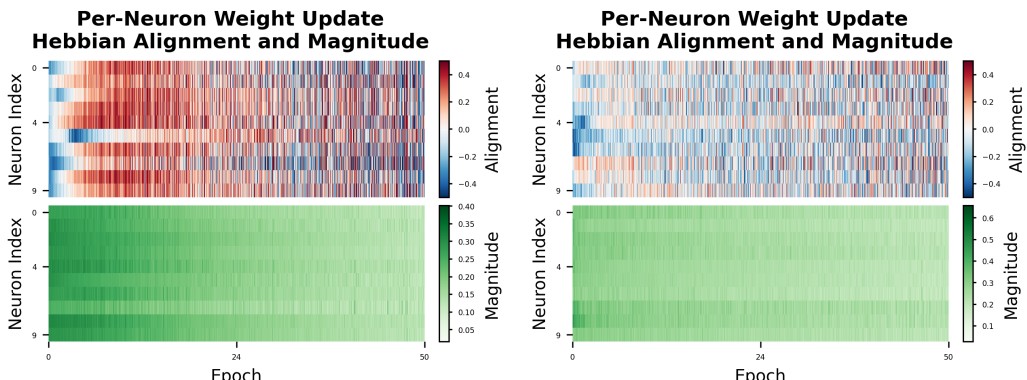

Figure 20: Early into training neurons often take on roles where their updates strongly bias towards Hebbian or Anti-Hebbian alignment. The diagram on the **left** depicts a SCE with ReLU activations, a 0.5x initialization scaling, a learning rate of 0.001 and a weight decay of 0.05. The diagram on the **right** depicts a similar SCE with ReLU activations and a learning rate of 0.01, but a lower weight decay of 0.025 and a default initialization.

Let $W_{i:}$ denote the $i$-th row of the weight matrix. Interpreting $i$ as the index of the neuron, this row can be seen as the synaptic efficacies of the synapses of this neuron. Here, we will allow every neuron to have a different weight decay. This is equivalent to the following generalized form of weight decay:

$$\frac{\gamma}{2}\operatorname{Tr}[W^T D W], \tag{20}$$

where $D$ is a diagonal positive-definite (PD) matrix. $D_{ii}$ is exactly the rate of decay for all the synapses of the $i$-th neuron. Of course, mathematically, this can be generalized a little bit further to allow different neurons to have correlated rates of decay, which could be biologically reasonable if these neurons are close in location. To achieve this, one simply has to allow $D$ to be a generic PD matrix, which includes the diagonal case as a special case.

For a generic learning rule $g(x, \theta)$ given in equation 21, the corresponding learning dynamics is:

$$\Delta W = g(x, \theta) - \gamma D W, \tag{21}$$

where $g$ is the learning rule and $\theta$ is the entirety of all trainable (plastic) parameters. For clarity, $\eta$ is subsumed into $g$ and $\gamma$. Close to stationarity, we have that $\mathbb{E}_x[g(x, \theta)] \approx \gamma D W$.

The direction of alignment at stationarity when $\mathbb{E}_x[g(x, \theta)] = \gamma W$ is thus

$$\operatorname{Tr}\left[\mathbb{E}_x[g(x, \theta)]\mathbb{E}_x[h_a h_b^T]\right] = \gamma \operatorname{Tr}\left[DW\mathbb{E}_x[h_a h_b^T]\right] \tag{22}$$

$$= \gamma \mathbb{E}[\|h_b\|_D^2] > 0, \tag{23}$$

where $\|h_b\|_D^2 = h_b^T D h_b$. We are done. Therefore, the theory extends naturally to the case when there is a nonuniform weight decay rate across neurons.

## C.5 Formal Theory

### C.5.1 Setup

We consider a single layer of a network. For each input $x$, let $h_a(x) \in \mathbb{R}^{n_a}$ denote the (post)activation of the previous layer and let

$$h_b(x; W) = W h_a(x) \in \mathbb{R}^{n_b}$$

be the preactivation of the current layer, where $W \in \mathbb{R}^{n_b \times n_a}$ is the weight matrix of this layer. We regard $h_a(x)$ as fixed and focus on this layer.

Let $\ell(x; W)$ be the loss on example $x$ and define the *penalized* expected loss

$$\mathcal{L}(W) := \mathbb{E}_x\big[\ell(x; W)\big] + \frac{\gamma}{2} \|W\|_F^2, \qquad \gamma > 0.$$

For a single example $x$, define the *learning signal* at this layer as the negative gradient with respect to $W$,

$$g(x; W) := -\nabla_W \ell(x; W) \in \mathbb{R}^{n_b \times n_a},$$

and the *Hebbian update* as

$$H(x; W) := h_b(x; W) h_a(x)^\top \in \mathbb{R}^{n_b \times n_a}.$$

We use the Frobenius inner product $\langle A, B \rangle := \mathrm{Tr}(A^\top B)$ on matrices.

We study the expected alignment

$$C(W) := \mathbb{E}_x\big[\, \langle g(x; W), H(x; W) \rangle \,\big].$$

A matrix $W^\star$ is called a *stationary point* if $\nabla\mathcal{L}(W^\star) = 0$, that is,

$$\mathbb{E}_x\big[\nabla_W \ell(x; W^\star)\big] + \gamma W^\star = 0 \qquad \Longleftrightarrow \qquad \mathbb{E}_x\big[g(x; W^\star)\big] = \gamma W^\star. \tag{24}$$

#### Structural and regularity assumptions

We make the following assumptions.

**Assumption 1** (Chain rule structure)**.** *For each $x$ and $W$, the loss is differentiable with respect to the preactivation $h_b$, and the gradient with respect to $W$ obeys the chain rule*

$$\nabla_W \ell(x; W) = \nabla_{h_b} \ell(x; W) h_a(x)^\top.$$

**Assumption 2** (Norm decoupling)**.** *There exists a constant $c_a > 0$ such that*

$$\|h_a(x)\|^2 = c_a \quad \text{for all } x$$

*almost surely with respect to the data distribution.*

Assumption 2 is a strong but simple way to encode the idea that the norm of the presynaptic activity is approximately constant, for example when activations are normalized.

**Assumption 3** (Local boundedness and Lipschitz continuity)**.** *There exist $r > 0$ and finite constants $M_g, M_H, L_g, L_H$ such that for all $x$ and all $W_1, W_2$ with $\|W_i - W^\star\|_F \leq r$, the following hold:*

$$\|g(x; W_i)\|_F \leq M_g, \qquad\qquad\qquad \|H(x; W_i)\|_F \leq M_H,$$

$$\|g(x; W_1) - g(x; W_2)\|_F \leq L_g \|W_1 - W_2\|_F, \quad \|H(x; W_1) - H(x; W_2)\|_F \leq L_H \|W_1 - W_2\|_F.$$

Assumption 3 is a local regularity condition on the layer dynamics. In concrete architectures it follows from bounded activations and Lipschitz nonlinearities in a neighborhood of $W^\star$.

### C.5.2 ALIGNMENT AT STATIONARITY

We first compute $C(W)$ at a stationary point $W^\star$ and show that it is strictly positive under the above assumptions.

**Lemma 1** (Value of $C(W)$ at stationarity). *Let $W^\star$ be a stationary point of $\mathcal{L}$ with $W^\star \neq 0$. Under Assumptions 1 and 2,*

$$C(W^\star) = \gamma\, c_a\, \|W^\star\|_F^2 \;>\; 0.$$

*Proof.* Fix $x$ and $W$. By Assumption 1,

$$g(x;W) = -\nabla_W \ell(x;W) = -\nabla_{h_b}\ell(x;W)\, h_a(x)^\top.$$

Recall also $H(x;W) = h_b(x;W)\, h_a(x)^\top$, where $h_b(x;W) = W h_a(x)$. Thus at $W^\star$ we have

$$g(x;W^\star) = -\nabla_{h_b}\ell(x;W^\star)\, h_a(x)^\top, \qquad H(x;W^\star) = h_b(x;W^\star)\, h_a(x)^\top.$$

The Frobenius inner product of these matrices is

$$\langle g(x;W^\star), H(x;W^\star)\rangle = \mathrm{Tr}\big(g(x;W^\star)^\top H(x;W^\star)\big)$$
$$= \mathrm{Tr}\Big(\big(-\nabla_{h_b}\ell(x;W^\star)\, h_a(x)^\top\big)^\top h_b(x;W^\star)\, h_a(x)^\top\Big)$$
$$= \mathrm{Tr}\big(-h_a(x)\,\nabla_{h_b}\ell(x;W^\star)^\top h_b(x;W^\star)\, h_a(x)^\top\big).$$

Pulling out the scalar $\nabla_{h_b}\ell(x;W^\star)^\top h_b(x;W^\star)$ and using $\mathrm{Tr}\big(h_a(x)\, h_a(x)^\top\big) = \|h_a(x)\|^2$ yields

$$\langle g(x;W^\star), H(x;W^\star)\rangle = -\|h_a(x)\|^2\, \nabla_{h_b}\ell(x;W^\star)^\top h_b(x;W^\star).$$

Taking expectations and using Assumption 2 ($\|h_a(x)\|^2 = c_a$ almost surely) gives

$$C(W^\star) = \mathbb{E}_x\big[\langle g(x;W^\star), H(x;W^\star)\rangle\big] = -c_a\, \mathbb{E}_x\big[\nabla_{h_b}\ell(x;W^\star)^\top h_b(x;W^\star)\big]. \qquad (25)$$

We next express the expectation on the right in terms of $W^\star$. By Assumption 1,

$$\nabla_W \ell(x;W^\star) = \nabla_{h_b}\ell(x;W^\star)\, h_a(x)^\top.$$

The gradient of $\mathcal{L}$ at $W^\star$ is

$$\nabla\mathcal{L}(W^\star) = \mathbb{E}_x\big[\nabla_W \ell(x;W^\star)\big] + \gamma W^\star = \mathbb{E}_x\big[\nabla_{h_b}\ell(x;W^\star)\, h_a(x)^\top\big] + \gamma W^\star.$$

By stationarity equation 24, this vanishes, hence

$$\mathbb{E}_x\big[\nabla_{h_b}\ell(x;W^\star)\, h_a(x)^\top\big] = -\gamma W^\star.$$

Let $g_b(x;W^\star) := \nabla_{h_b}\ell(x;W^\star)$ and $a(x) := h_a(x)$ for brevity. Then

$$\mathbb{E}_x\big[g_b(x;W^\star)\, a(x)^\top\big] = -\gamma W^\star.$$

Take the Frobenius inner product of both sides with $W^\star$:

$$\big\langle \mathbb{E}_x[g_b a^\top], W^\star\big\rangle = \langle -\gamma W^\star, W^\star\rangle = -\gamma\, \|W^\star\|_F^2.$$

The left side can be rewritten as

$$\big\langle \mathbb{E}_x[g_b a^\top], W^\star\big\rangle = \mathrm{Tr}\big(\mathbb{E}_x[g_b a^\top]^\top W^\star\big) = \mathrm{Tr}\big(\mathbb{E}_x[a g_b^\top]\, W^\star\big) = \mathbb{E}_x\big[\mathrm{Tr}(a g_b^\top W^\star)\big].$$

Using $h_b(x;W^\star) = W^\star a(x)$,

$$g_b(x;W^\star)^\top h_b(x;W^\star) = g_b(x;W^\star)^\top W^\star a(x) = \mathrm{Tr}\big(a(x)\, g_b(x;W^\star)^\top W^\star\big),$$

so that

$$\mathbb{E}_x\big[g_b(x;W^\star)^\top h_b(x;W^\star)\big] = \mathbb{E}_x\big[\mathrm{Tr}(a g_b^\top W^\star)\big] = \big\langle \mathbb{E}_x[g_b a^\top], W^\star\big\rangle.$$

We conclude that

$$\mathbb{E}_x\big[\nabla_{h_b}\ell(x;W^\star)^\top h_b(x;W^\star)\big] = -\gamma\, \|W^\star\|_F^2. \qquad (26)$$

Substituting equation 26 into equation 25 yields

$$C(W^\star) = -c_a\big(-\gamma\, \|W^\star\|_F^2\big) = \gamma\, c_a\, \|W^\star\|_F^2.$$

Since $\gamma > 0$, $c_a > 0$, and $W^\star \neq 0$, this is strictly positive. $\qquad\square$

Lemma 1 shows that at any nonzero stationary point the learning signal is positively aligned with the Hebbian update, formally recovering equation 8.

### C.5.3 A LOCAL BOUND OUT-OF-STATIONARITY

We now show that this positive alignment persists in a full neighborhood of $W^\star$, and we give an explicit quantitative bound in terms of the distance to $W^\star$.

**Lemma 2** (Lipschitz continuity of $C(W)$ near $W^\star$). *Under Assumption 3, there exists a constant*

$$L_C := L_g M_H + M_g L_H$$

*such that for all $W_1, W_2$ with $\|W_i - W^\star\|_F \le r$,*

$$\left| C(W_1) - C(W_2) \right| \le L_C \left\| W_1 - W_2 \right\|_F.$$

*Proof.* Fix $W_1, W_2$ with $\|W_i - W^\star\|_F \le r$. For each $x$, abbreviate $g_i(x) := g(x; W_i)$ and $H_i(x) := H(x; W_i)$. Then

$$C(W_1) - C(W_2) = \mathbb{E}_x\big[\,\langle g_1(x), H_1(x)\rangle - \langle g_2(x), H_2(x)\rangle\,\big]$$
$$= \mathbb{E}_x\big[\,\langle g_1(x) - g_2(x), H_1(x)\rangle + \langle g_2(x), H_1(x) - H_2(x)\rangle\,\big].$$

Taking absolute values and applying the Cauchy–Schwarz inequality,

$$\left| C(W_1) - C(W_2) \right| \le \mathbb{E}_x\big[\,\|g_1(x) - g_2(x)\|_F\,\|H_1(x)\|_F + \|g_2(x)\|_F\,\|H_1(x) - H_2(x)\|_F\,\big].$$

By Assumption 3,

$$\|g_1(x) - g_2(x)\|_F \le L_g\,\|W_1 - W_2\|_F, \qquad \|H_1(x) - H_2(x)\|_F \le L_H\,\|W_1 - W_2\|_F,$$

and

$$\|g_2(x)\|_F \le M_g, \qquad \|H_1(x)\|_F \le M_H.$$

Hence

$$\left| C(W_1) - C(W_2) \right| \le \mathbb{E}_x\Big[L_g\,\|W_1 - W_2\|_F\,M_H + M_g L_H\,\|W_1 - W_2\|_F\Big]$$
$$= \big(L_g M_H + M_g L_H\big)\,\|W_1 - W_2\|_F.$$

This proves the claim. $\qquad\square$

Combining Lemma 1 with Lemma 2 gives the desired near-stationary bound.

**Theorem 1** (Hebbian alignment in a neighborhood of stationarity). *Let $W^\star$ be a stationary point of $\mathcal{L}$ with $W^\star \ne 0$. Assume Assumptions 1, 2, and 3. Set*

$$C_\star := C(W^\star) = \gamma c_a \|W^\star\|_F^2 > 0, \qquad L_C := L_g M_H + M_g L_H.$$

*Then for every $W$ such that $\|W - W^\star\|_F \le r$,*

$$C(W) \ge C_\star - L_C\,\|W - W^\star\|_F. \tag{27}$$

*In particular, if*

$$\|W - W^\star\|_F \le \frac{C_\star}{2L_C},$$

*then*

$$C(W) \ge \frac{C_\star}{2} > 0.$$

*Proof.* The inequality equation 27 follows directly from Lemma 2 with $W_1 = W$, $W_2 = W^\star$:

$$\left| C(W) - C_\star \right| = \left| C(W) - C(W^\star) \right| \le L_C\,\|W - W^\star\|_F.$$

Rearranging yields

$$C(W) \ge C_\star - L_C\,\|W - W^\star\|_F.$$

If $\|W - W^\star\|_F \le C_\star/(2L_C)$, then

$$C(W) \ge C_\star - L_C\frac{C_\star}{2L_C} = \frac{C_\star}{2} > 0.$$

$\qquad\square$

Theorem 1 states that once the weights enter a neighborhood of a nonzero stationary point, the expected alignment between the learning signal and the Hebbian update is bounded away from zero and remains strictly positive, with a margin that decreases at most linearly with the distance from $W^\star$. This already implies that if the learning update is upper bounded by $v$, then for at least

$$\tau = \frac{r}{v} \tag{28}$$

duration of time, the learning dynamics must have a positive Hebbian alignment. Of course, in reality, the closer one gets to the stationary point, the slower the dynamics becomes, and so the time in reality could be infinitely long (as in a linear dynamics, where the dynamical variable only reaches the stationary point at the infinite-time limit).

It could also be desirable to directly link this distance to the dynamics, which we achieve in the next section.

### C.5.4 A BOUND IN TERMS OF THE STATIONARITY GAP

As an alternative perspetive, it could be desirable to express the proximity to $W^\star$ not by $\lVert W - W^\star \rVert_F$ but by the size of the drift of the continuous-time gradient flow.

Define the drift field

$$F(W) \coloneqq \mathbb{E}_x\big[g(x; W)\big] - \gamma W = -\mathbb{E}_x\big[\nabla_W \ell(x; W)\big] - \gamma W.$$

Then the gradient flow associated with $\mathcal{L}$ is given by

$$\dot{W}(t) = F(W(t)).$$

The stationary points of $\mathcal{L}$ are the zeros of $F$; in particular $F(W^\star) = 0$. We now assume that $F$ is differentiable and that $W^\star$ is a nondegenerate zero of $F$, meaning that the Jacobian $DF(W^\star)$ is invertible. Invertibility implies a quantitative relation between the distance to $W^\star$ and the magnitude of $F(W)$.

**Assumption 4** (Nondegenerate stationary point)**.** *The mapping $F$ is continuously differentiable in a neighborhood of $W^\star$, and the Jacobian $DF(W^\star)$ is invertible. Denote by $\sigma_{\min} > 0$ the smallest singular value of $DF(W^\star)$.*

**Lemma 3** (Control of distance by vector field)**.** *Under Assumption 4, there exist $r_0 > 0$ and $\mu > 0$ such that for all $W$ with $\lVert W - W^\star \rVert_F \leq r_0$,*

$$\lVert F(W) \rVert_F \geq \mu \lVert W - W^\star \rVert_F.$$

*In particular one may take $\mu = \sigma_{\min}/2$ for $r_0$ small enough.*

*Proof.* By the mean value formula in Banach spaces,

$$F(W) - F(W^\star) = \int_0^1 DF\big(W^\star + t(W - W^\star)\big)(W - W^\star)\, dt.$$

Since $F(W^\star) = 0$, we have

$$F(W) = \int_0^1 DF\big(W^\star + t(W - W^\star)\big)(W - W^\star)\, dt.$$

Fix $\varepsilon \in (0, 1)$. By continuity of $DF$ and invertibility of $DF(W^\star)$, there exists $r_0 > 0$ such that for all $W$ with $\lVert W - W^\star \rVert_F \leq r_0$ and all $t \in [0, 1]$, the smallest singular value of $DF(W^\star + t(W - W^\star))$ is at least $(1 - \varepsilon)\sigma_{\min}$. In particular, the operator norm of the inverse of each such Jacobian is at most $[(1 - \varepsilon)\sigma_{\min}]^{-1}$.

For any $v$,

$$\big\lVert DF\big(W^\star + t(W - W^\star)\big)v \big\rVert_F \geq (1 - \varepsilon)\sigma_{\min} \lVert v \rVert_F.$$

Applying this with $v = W - W^\star$ and using Minkowski's inequality gives

$$\|F(W)\|_F = \left\| \int_0^1 DF\big(W^\star + t(W - W^\star)\big)(W - W^\star)\, dt \right\|_F$$

$$\geq \int_0^1 \left\| DF\big(W^\star + t(W - W^\star)\big)(W - W^\star) \right\|_F dt$$

$$\geq \int_0^1 (1 - \varepsilon)\sigma_{\min} \|W - W^\star\|_F\, dt$$

$$= (1 - \varepsilon)\sigma_{\min} \|W - W^\star\|_F.$$

Choosing $\varepsilon = 1/2$ and setting $\mu = \sigma_{\min}/2$ yields the stated inequality. $\qquad\square$

We can now express the Hebbian alignment bound in terms of the stationarity gap $\|F(W)\|_F$.

**Theorem 2** (Hebbian bound in terms of stationarity gap). *Assume the hypotheses of Theorem 1 and Assumption 4. Let $\mu > 0$ and $r_0 > 0$ be as in Lemma 3. Then there exists $r' > 0$ such that for all $W$ with $\|W - W^\star\|_F \leq r'$,*

$$C(W) \geq C_\star - \frac{L_C}{\mu} \|F(W)\|_F. \tag{29}$$

*In particular, if*

$$\|F(W)\|_F \leq \frac{\mu C_\star}{2 L_C},$$

*then $C(W) \geq C_\star/2 > 0$.*

*Proof.* Let $r' := \min\{r, r_0\}$, where $r$ is from Assumption 3 and $r_0$ from Lemma 3. For any $W$ with $\|W - W^\star\|_F \leq r'$, Theorem 1 gives

$$C(W) \geq C_\star - L_C \|W - W^\star\|_F.$$

By Lemma 3, we have $\|W - W^\star\|_F \leq \mu^{-1} \|F(W)\|_F$, hence

$$C(W) \geq C_\star - L_C \frac{1}{\mu} \|F(W)\|_F,$$

which is equation 29. If in addition $\|F(W)\|_F \leq \mu C_\star/(2L_C)$, then

$$C(W) \geq C_\star - L_C \frac{1}{\mu} \frac{\mu C_\star}{2 L_C} = \frac{C_\star}{2} > 0.$$

$\qquad\square$

Theorem 2 formalizes the intuitive statement that *once the expected update field $F(W)$ is small, the learning signal remains strongly Hebbian.* In particular, as $W$ approaches a nondegenerate stationary point of the penalized loss, the expected alignment between $g(x; W)$ and $H(x; W)$ stays bounded away from zero, with a deficit that scales at most linearly in the norm of the stationarity gap $\|F(W)\|_F$.

### C.6    RANDOMNN FORMULATION

The RandomNN was a MLP with 3 hidden vectors of size 128 and tanh activations. The MLP took the same input as the student model but outputted a vector of length 4. The output was averaged across the batch and then and then multiplied by a random projection matrix unique to each parameter and reshaped to be the dimensions of that parameter. No parameters of RandomNN change after initialization. The resulting learning signal for $W$ is a deterministically random low-rank matrix, $W^\star$.

The full weight update is given by:

$$\Delta W = \eta \big( g(x, \theta) - \gamma W \big)$$

where

$$g(x, \theta) = p(W^*) s_{dir}(W) s_{red}(W, W^*) W^*$$

and where,

$$s_{red}(W, W^*) = \text{sign}\left(\|W\|_2 - \|W - W^*\|_2\right)$$

$$s_{dir}(W) = \text{sign}\left(100 - \|W\|_2\right)$$

$$p(W^*) = \begin{cases} 1 & \text{if } \|W^*\|_2 \leq 1 \\ \frac{1}{\|W^*\|_2 + \epsilon} & \text{otherwise} \end{cases}$$

The minimal requirements to have non-zero weights and reach stationarity require $g(x, \theta)$ to be some forcing function that wants to make the weights larger than zero. This is the case with any descent learning algorithm, as with zero weights, one can not learn or express anything besides 0. However, it is not only true of learning algorithms.

There are a number of trivial constructions that satisfy this condition, such as setting $f(x, \theta) = A$ where $A$ is a random matrix defined at initialization. This will naturally be an expanding force and become aligned with the Hebbian rule, however, it will do this even without regularization. But is there a way to make a non-learning model that does not behave Hebbian at all without regularization, but does with regularization?

RandomNN is one such construction. In it, we produce random weight update vectors in a subspace of the possible directions of the student model's weight updates. This means that after some number of updates, the value of the weight is not orthogonal to the random update vectors, and in fact becomes highly aligned to them. Thus, for a given weight update, the norm of the weights will either increase or decrease, not strictly increase. We can make an attractor to push the norm of the weights to a specific non-zero value by choosing to either add or subtract the random update, depending on which one will move it closer to the target value. Thus, without any regularization, the model's weights will converge to have the target norm and will, on average, not increase or decrease, resulting in no Hebbian alignment. However, once a weight decay term is added, the attractor will try to strictly increase to approach the target, and thus align with the Hebbian update. We also apply a weight update norm clip for stability.

