# OpenReview forum: "Ubiquity of Hebbian Dynamics in Complex Learning Rules"
_ICLR.cc/2026/Conference — Submitted to ICLR 2026_

### Official Review · Reviewer_td9e · 2025-10-26

**Soundness:** 4
**Presentation:** 3
**Contribution:** 3
**Rating:** 8
**Confidence:** 3

**Summary:**

There is wide empirical evidence for Hebbian learning in the brain. This paper challenges this by showing that strong correlation with Hebbian learning can be achieved by many learning rules that have, in principle, nothing to do with Hebbian learning, as long as they are coupled with weight decay. Additionally, it shows noise on the parameters makes the learning signal more anti-Hebbian. The claims are supported by heuristic theoretical derivations at stationarity as well as experiments on relatively small neural networks.

**Strengths:**

The arguments presented in the paper are simple yet powerful, and raise significant questions regarding our understanding of synaptic plasticity in the brain. The theoretical arguments provide solid intuition that is shown to hold in practice through careful experimentation. The discussion related to neuroscience is well done, and I had the pleasant surprise of finding all the questions the paper raised during my first read carefully addressed.

**Weaknesses:**

What follows should be considered as suggestions rather than weaknesses:

- **Figure 4 could be moved to the appendix.** The results it shows are very intuitive (as mentioned in the main text), and the setup differs from the rest of the paper, requiring considerable time to understand what is happening. As a result, reading this figure alone may bring more confusion than clarity.
- **The analysis in Section 4 could be made more precise**. The setup considered in this section is simple enough that closed-form solutions of the learning dynamics can be derived (see e.g., Saxe et al. 2013), allowing the evolution of alignment with the Hebbian update to be precisely characterized. This could help better understand the alignment dynamics.
- **The connection to empirical evidence could be strengthened.** The paper shows that traces of Hebbian learning do not necessarily imply that the underlying learning rule is Hebbian, without directly commenting on whether current empirical evidence suffers from this problem. The following experiment would help: reproduce the methodology of empirical neuroscience papers establishing the existence of Hebbian learning in the brain that the authors cite, and show that the same results can be achieved by a version of SGD. I suspect this would be possible given the current results of the paper, and having this experiment would help make the argument that current evidence is insufficient to conclude that brains learn with Hebbian learning.

**Questions:**

-

---

> ### Author Response · Authors · 2025-11-22
>
> We appreciate your very positive evaluation and your suggestions. Below, we address each point.
>
> **S1:** Figure 4 in the main text.
> > Figure 4 could be moved to the appendix; its setup differs from the rest of the paper and may confuse more than help.
>
> Thank you for this suggestion. We added this figure to clarify how Hebbian learning alone doesn’t reproduce SGD-like features on the tasks we were evaluating on; however, we do see how the setup could be confusing. In the revision, we will move this figure to the appendix and tighten its description there.
>
> **S2:** Precision of Section 4 dynamics
> > The setup in Section 4 is simple enough to allow closed-form learning dynamics (e.g., Saxe et al. 2013), which could clarify how alignment evolves.
>
> Our original choice was to emphasize the stationary analysis because it is simple and general. That said, after your suggestion and those of some of the other reviewers, we added an out-of-stationarity analysis in Appendix C5.
>
> **S3:** Stronger link to empirical Hebbian evidence
> > The connection to empirical evidence could be strengthened. The paper shows that traces of Hebbian learning do not necessarily imply that the underlying learning rule is Hebbian, without directly commenting on whether current empirical evidence suffers from this problem. The following experiment would help: reproduce the methodology of empirical neuroscience papers establishing the existence of Hebbian learning in the brain that the authors cite, and show that the same results can be achieved by a version of SGD. I suspect this would be possible given the current results of the paper, and having this experiment would help make the argument that current evidence is insufficient to conclude that brains learn with Hebbian learning.
>
> To emphasize, our work is about presenting an alternative explanation of Hebbian learning to the well-established functional one as described in the Summary Rebuttal. However, we do agree that this could contribute to strengthening our paper. We did make an attempt to do this during the rebuttal period, but unfortunately, we did not have time to implement such a model. We will consider doing this in the final version. At the end of the day, this theory would most strongly benefit from wet lab experimental support, and any modeling we do would be an imperfect reproduction of that, limiting what conclusion we could make.

---

> ### Comment · Reviewer_td9e · 2025-11-22
>
> I thank the authors for their detailed rebuttal and for the numerous updates they made to enrich and nuance their claims.
>
> I agree that evidence from wet labs would ultimately be valuable. However, I would still argue that providing alternative hypotheses that quantitatively explain existing results is precisely what motivates these labs to pursue such experiments in the first place.
>
> Overall, I am convinced that this is a **very strong paper that should be highlighted** at the conference (I have therefore updated my score from 8 to 10). I must say that I am quite surprised by the negative feedback from other reviewers.
>
> As a disclaimer, I am only superficially familiar with the Hebbian learning literature. However, as the authors highlight in their rebuttal, their claims are not about better understanding Hebbian learning per se, but rather about demonstrating that Hebbian learning traces can be found across many learning rules. In other words, their results are largely orthogonal to our understanding of Hebbian learning itself.

---

### Official Review · Reviewer_7Qh1 · 2025-10-29

**Soundness:** 3
**Presentation:** 3
**Contribution:** 2
**Rating:** 4
**Confidence:** 3

**Summary:**

The authors show that in the presence of weight decay, the "learning signal" caused by SGD will tend to align with Hebbian updates (input time output), whereas in the presence of high weight noise it will instead tend to align with anti-Hebbian updates.

Several experiments confirm the intuitive results.

**Strengths:**

The general approach of predicting which conditions will align learning with Hebbian or anti-Hebbian updates seems novel, to my knowledge.  The experiments, if correct, seem to confirm the proposals.

Some of the results are genuinely unexpected, e.g. the neat quadratic boundary shown in Figure 5.

**Weaknesses:**

The main problem is that the paper consistently presents itself as an explanation of *why* Hebbian/anti-Hebbian learning occurs, while being based entirely on homeostatic mechanisms and completely ignoring functional outcomes of Hebbian learning.

For example, Hebbian learning is a powerful pattern learner, and anti-Hebbian learning enforces decorrelation (a famous example of both together is Foldiak 1991 https://pubmed.ncbi.nlm.nih.gov/2291903/ ).

Since (as the authors acknowledge) Hebbian updates will tend to be aligned with the weight vectors (at least after a certain amount of learning) and thus increase W magnitude, then keeping weights stationary in the face of weight decay basically requires some kind of alignment with Hebbian updates.

To make this obvious homeostatic necessity a general principle, you first need to assume that something like uniform weight decay occurs in the brain, which is not at all obvious (e.g. I understand daily turnover of synapses tends to erase weak synapses but preserve strong ones). But even if it did, it would not follow that this should explain all, or even most, of Hebbian/anti-Hebbian learning in the brain, which the paper basically hints at repeatedly.

Similarly, observing anti-Hebbian updates in the presence of noise would not exactly confirm the theory, since this is precisely the expected outcome of several well-known Hebbian learning rules, including BCM and most STDP rules (which involve larer negative than positive windows). The reason for this is precisely to degrade synapses between neurons that fire uncorrelatedly.

Thus, while some intriguing results are reported, the paper needs to be significantly "toned down" and clarify the reach of its proposals.

**Questions:**

- Eq.4: please add some more parentheses, or spaces - it's not immediately clear what is included in the gradient sign (I think it is just the gradient of l, but please clarify)

- Please expand the maths  between eq 4 and eq 5 (maybe in an appendix). How do you get from 4 to 5 ? Why did the h_a become a second h_b? Why are there traces involved?

- In equation 9, what exactly is theta, and how does it differ from W? Why "cosine similarity between the learning rule and the Hebbian rule" is somehow cos Theta? And, again, where do the traces come from?

- Please explain Eq 17 a bit more. What are c0 and c1, and are they really "constants"?

- Where's the data to support the next-to-last paragraph in p. 8 (line 420-424)?

- In Figure 7, what does "Init: 0.5x", etc. mean?

- I note that the Hebbian update is simply the gradient of y^2, so the initial bump of Hebbian-ness in Figure 7 may be caused by the need to increase response outputs, regardless of overall W magnitude. Perhaps plotting y (output) magnitude would be useful.

- Minor: fix the parentheses in line 131.

---

> ### Author Response · Authors · 2025-11-22
>
> Thank you for your review of our paper, and we appreciate your highlighting the novelty of our findings. We believe most of your criticisms are due to a misunderstanding of the scope of our work (see the Summary Rebuttal). We make two claims in our conclusion:
>
> C1: “*First, our results show that Hebbian and anti-Hebbian plasticity can emerge as regimes of gradient-based optimization, **in addition to** their conventional role as fundamental learning mechanisms.”*
>
> C2: “Second, the presence of Hebbian or anti-Hebbian signatures in neuro-physiological data **need not be interpreted as evidence against** global error-driven optimization in the brain.*”
>
> With this context, we address your concerns below.
>
> **W1:** "The main problem is that the paper consistently presents itself as an explanation of why Hebbian/anti-Hebbian learning occurs, while being based entirely on homeostatic mechanisms and completely ignoring functional outcomes of Hebbian learning."
>
> This is a misunderstanding of our claim and contribution. See our quote at the beginning. We explicitly said that our theory offers a **new** mechanism “**in addition to**” conventional ones, **not** to replace the conventional ones.
>
> To emphasize, our work is about presenting an alternative **explanation** of Hebbian learning to the well-established functional one:
>
> 1. Conventional-Functional: functional Hebbian learning, fire together, wire together.
> 2. Emergent: Any regularized learning signal
>
> As we explain in the conclusion, the homeostatic mechanism could broadly coexist with the conventional functional one. We have edited the abstract and introduction to make this scope even more explicit and to avoid language that suggests we explain all Hebbian or anti-Hebbian learning in the brain.
>
> **W2:** ...you first need to assume that something like uniform weight decay occurs in the brain, which is not at all obvious.
>
> Exploring all non-uniform variants of L2 weight decay is too broad for thorough empirical testing, but Appendix C.4 extends our analysis to this domain. We’ve added a note on this in the Neurobiology part of Section 3.
>
> **W3:** ...observing anti-Hebbian updates in the presence of noise would not exactly confirm the theory, since this is precisely the expected outcome of several well-known Hebbian learning rules, including BCM and most STDP rules...
>
> We believe this question comes from the misunderstanding of our work we identified above; this criticism is exactly one of the key insights offered by our work: observations of Hebbian/Anti-Hebbian mechanisms are insufficient to confirm **any** (especially the Hebbian ones) previously proposed mechanism. This has been clear from our original conclusion, see C1 and C2 quoted at the beginning. Additionally, our theory also predicts a quadratic phase boundary when adding noise, which could serve as an experimental validation.
>
> We never claimed that the proposed mechanism is the correct one, but that the field could be ignoring an alternative explanation, and so future experimental works should devise methods to tell them part (namely, the conventional vs emergent Hebbianity). We will revise the related work section to explicitly connect our findings to BCM and STDP-based anti-Hebbian effects and to emphasize that our mechanism is one possible way to generate similar signatures, not a replacement for those models.
>
> Q1: Thank you for this suggestion. We have clarified in the revision.
>
> Q2: We have made clarifications in the derivation. Thank you.
> Q3: We have clarified the notion in the revision. $cos(\theta)$ just represents the cosine similarity. Please let us know if you feel there should be further changes.
>
> Q4: We added a brief explanation for c1 and c2 in the revision to clarify that they are constants with respect to weight decay and noise.
>
> Q5: We added Figure 20 to the Appendix to show what we are describing.
>
> Q6: We updated the figure description to clarify that they mean a constant scaling of the original initialization.
>
> Q7: This is a good idea and likely is the case; we will do this for the final revision.
>
> Q8: Fixed. Thank you.

---

> > ### Comment · Reviewer_7Qh1 · 2025-11-25
> >
> > > observations of Hebbian/Anti-Hebbian mechanisms are insufficient to confirm any (especially the Hebbian ones) previously proposed mechanism.
> >
> > I'm sorry, what does that mean? Observations of Hebbian mechanisms do not confirm Hebbian mechanisms?
> >
> > Experiments do not observe a residual-from-L2-loss signal. They observe the actual **total** weight change, resulting from a certain input and a certain output.
> >
> > These inputs and outputs are often controlled by direct electrical stimulation of single neurons and axons, independently of any external stimulus, much less "global learning signals". The resulting weight change is (sometimes) Hebbian. In some cases, the mechanisms underlying the weight changes have been elucidated at the molecular level. That *is* the mechanism.
> >
> > The fact that roughly-Hebbian-aligned changes would be necessary to counteract a *hypothetical* weight decay, in the long run, does not seem relevant to such controlled neurophysiology experiments?
> >
> > IMO the authors need to clarify their thinking (which seems rather obscure at the moment, as illustrated by the sentence quoted above), make a more precise argument, and then submit that again.

---

> ### Author Response · Authors · 2025-11-25
>
> We thank the reviewer for the careful follow-up and for prompting us to clarify our wording and biological assumptions.
>
> > “observations of Hebbian/Anti-Hebbian mechanisms are insufficient to confirm any (especially the Hebbian ones) previously proposed mechanism.” I'm sorry, what does that mean? Observations of Hebbian mechanisms do not confirm Hebbian mechanisms?
> >
>
> The sentence you quote contains a typo: the first “mechanisms” should have read “phenomenology” (i.e. correlation to a Hebbian update). Our intended claim is that *observations of Hebbian or anti-Hebbian phenomenology do not uniquely identify the underlying learning algorithm*. We agree that experiments robustly demonstrate Hebbian- and STDP-like plasticity at identified synapses (e.g. Caporale and Dan, 2008; Feldman, 2012), often with well-characterized molecular pathways.
>
> > Experiments do not observe a residual-from-L2-loss signal. They observe the actual **total** weight change, resulting from a certain input and a certain output. These inputs and outputs are often controlled by direct electrical stimulation of single neurons and axons, independently of any external stimulus, much less "global learning signals". The resulting weight change is (sometimes) Hebbian. In some cases, the mechanisms underlying the weight changes have been elucidated at the molecular level. That *is* the mechanism. The fact that roughly-Hebbian-aligned changes would be necessary to counteract a *hypothetical* weight decay, in the long run, does not seem relevant to such controlled neurophysiology experiments?
> >
>
> We are absolutely in agreement that there have been very detailed and well controlled experiments that have shown the existence of local Hebbian computation. Importantly, it is also well established that compensatory processes are required to avoid runaway dynamics and maintain stable firing rates, consistent with work on the interaction between Hebbian and homeostatic plasticity (Turrigiano, 2012; Keck et al., 2017). As emphasized by Zenke and Gerstner (2017), Hebbian-like plasticity in recurrent networks **generally requires** **weight dependent compensatory processes that operate on a different, often slower, time scale**.
>
> The **time scales** for the learning signal and decay **do not need to be the same**, so the “**full weight update”** is not solely captured by the immediate response to the stimuli.
>
> In our framework, the **L2 term is a minimal model** of such contractive, weight dependent influences, but the analysis and central conclusion extend to any regularizer that imposes a contractive effect on large weights. Several models already incorporate this separation of influences, including voltage based STDP with homeostasis (Clopath et al., 2010), factorized Hebbian–homeostatic rules (Toyoizumi et al., 2014), and synaptic competition for a limited pool of building blocks (Triesch et al., 2018). Our findings suggest that, when the learning signal and decay process are distinct, an additional emergent Hebbian-like influence appears even if the learning signal is not functionally Hebbian.
>
> This influence is not tied to any specific learning rule, though it can mimic Hebbian learning over different time scales, since decay operates at a slower time scale than growth. The presence of Hebbian-style functional rules does not rule out more complex heterosynaptic mechanisms in the brain. This is, of course, what was intended by the phrase “global,” not that the brain literally optimizes a scalar objective. In fact, there are theoretical proposals with varying amounts of support for a great many learning signals in various parts of the brain that are not purely Hebbian (Chistiakova et al., 2014, 2015; Chater and Goda, 2021; Frey and Morris, 1997; Redondo and Morris, 2011; Losonczy et al., 2008; Govindarajan et al., 2011; Kastellakis et al., 2015; Oh et al., 2015; Perea and Araque, 2007; Gordon et al., 2009; Navarrete et al., 2012; De Pittà et al., 2016; Yagishita et al., 2014; Frémaux and Gerstner, 2016; Kuśmierz et al., 2017; Brzosko et al., 2019; Gerstner et al., 2018).
>
> Our manuscript does not provide a functional model of Hebbian computation. Rather, it proposes a theory and provides empirical support to show that so long as there exist the well-established decay mechanisms in the brain, data supporting other learning rules might be obscured by the emergent Hebbian learning. This is a cautionary point for neuroscientists.

---

> > ### Author Response · Authors · 2025-11-25
> >
> > > IMO the authors need to clarify their thinking (which seems rather obscure at the moment, as illustrated by the sentence quoted above), make a more precise argument, and then submit that again.
> > >
> >
> > The first point is an important clarification in our original reply, though a purely linguistic one and **does not materially detract** from the ideas we present in our work. The final point you raise, however, is a very good one! We do agree there is substantial experimental evidence in the brain for Hebbian learning. This is why in our original conclusion we specified “Given that unsupervised adaptation and reinforcement are useful and widespread mechanisms, intrinsically Hebbian homosynaptic plasticity likely does exist in the brain” and further emphasized this point the in the revised abstract: “our proposed mechanisms do not rule out any conventionally established forms of Hebbian plasticity and could coexist with them…”
> >
> > We struggle to see exactly what is obscure about our argument, especially given the revision in which we worked hard to clarify a number of points raised by reviewers. We will update the conclusion to further clarify these points.
> >
> > Citations:
> >
> > Brzosko et al., 2019: Neuromodulation of spike-timing-dependent plasticity: Past, present, and future. Neuron.
> > Caporale and Dan, 2008: Spike timing-dependent plasticity: A Hebbian learning rule. Annual Review of Neuroscience.
> > Chater and Goda, 2021: My neighbour hetero: Deconstructing the mechanisms underlying heterosynaptic plasticity. Current Opinion in Neurobiology.
> > Chistiakova et al., 2014: Heterosynaptic plasticity: Multiple mechanisms and multiple roles. The Neuroscientist.
> > Chistiakova et al., 2015: Homeostatic role of heterosynaptic plasticity: Models and experiments. Frontiers in Computational Neuroscience.
> > Clopath et al., 2010: Connectivity reflects coding: A model of voltage-based spike-timing-dependent plasticity with homeostasis. Nature Neuroscience.
> > De Pittà et al., 2016: Astrocytes: Orchestrating synaptic plasticity? Neuroscience.
> > Feldman, 2012: The spike-timing dependence of plasticity. Neuron.
> > Frey and Morris, 1997: Synaptic tagging and long-term potentiation. Nature.
> > Frémaux and Gerstner, 2016: Neuromodulated spike-timing-dependent plasticity, and theory of three-factor learning rules. Frontiers in Neural Circuits.
> > Gerstner et al., 2018: Eligibility traces and plasticity on behavioral time scales: Experimental support of neoHebbian three-factor learning rules. Frontiers in Neural Circuits.
> > Gordon et al., 2009: Astrocyte-mediated distributed plasticity at hypothalamic glutamate synapses. Neuron.
> > Govindarajan et al., 2011: The dendritic branch is the preferred integrative unit for protein synthesis-dependent LTP. Neuron.
> > Kastellakis et al., 2015: Synaptic clustering within dendrites: An emerging theory of memory formation. Progress in Neurobiology.
> > Keck et al., 2017: Integrating Hebbian and homeostatic plasticity: The current state of the field and future research directions. Philosophical Transactions of the Royal Society B: Biological Sciences.
> > Kuśmierz et al., 2017: Learning with three factors: Modulating Hebbian plasticity with errors. Current Opinion in Neurobiology.
> > Losonczy et al., 2008: Compartmentalized dendritic plasticity and input feature storage in neurons. Nature.
> > Navarrete et al., 2012: Astrocytes mediate in vivo cholinergic-induced synaptic plasticity. PLoS Biology.
> > Oh et al., 2015: Heterosynaptic structural plasticity on local dendritic segments of hippocampal CA1 neurons. Cell Reports.
> > Perea and Araque, 2007: Astrocytes potentiate transmitter release at single hippocampal synapses. Science.
> > Redondo and Morris, 2011: Making memories last: The synaptic tagging and capture hypothesis. Nature Reviews Neuroscience.
> > Toyoizumi et al., 2014: Modeling the dynamic interaction of Hebbian and homeostatic plasticity. Neuron.
> > Triesch et al., 2018: Competition for synaptic building blocks shapes synaptic plasticity. eLife.
> > Turrigiano, 2012: Homeostatic synaptic plasticity: Local and global mechanisms for stabilizing neuronal function. Cold Spring Harbor Perspectives in Biology.
> > Yagishita et al., 2014: A critical time window for dopamine actions on the structural plasticity of dendritic spines. Science.
> > Zenke and Gerstner, 2017: Hebbian plasticity requires compensatory processes on multiple timescales. Philosophical Transactions of the Royal Society B: Biological Sciences.

---

### Official Review · Reviewer_GwC8 · 2025-10-31

**Soundness:** 1
**Presentation:** 2
**Contribution:** 1
**Rating:** 2
**Confidence:** 5

**Summary:**

The paper claims that Hebbian and anti-Hebbian plasticity emerge universally from gradient-based optimization methods when combined with L2 weight decay or noise. The authors derive analytic results showing that learning rules with L2 regularization align with Hebbian updates near stationarity, while stochastic noise induces anti-Hebbian alignment. They support their theory with experiments on small MLPs and Transformers, suggesting that observed Hebbian/anti-Hebbian plasticity in the brain might be a signature of general optimization processes.

**Strengths:**

- The paper is clearly written and structured.
- The attempt to bridge biological learning and gradient-based optimization is an important unsolved question.
- The experiments provide systematic and broad qualitative confirmation of the claimed effects.

**Weaknesses:**

The following weaknesses highlight fundamental conceptual and theoretical issues that undermine the relevance of the paper’s main claims, distinguishing between superficial equilibrium effects and genuine learning dynamics.
- The paper equates the norm-stabilizing effect of L2 regularization with Hebbian dynamics. L2 weight decay enforces contraction of weight norms, while Hebbian learning refers to directional correlation between pre- and post-synaptic activity (e.g., feature extraction). The alignment the authors observe is a trivial consequence of regularization equilibrium (($\nabla_\theta \ell + \gamma W = 0$)), not evidence of Hebbian computation.
- Classical linear Hebbian rules, when combined with weight normalization or decay, are mathematically equivalent to stochastic gradient descent on the PCA objective (Oja, 1982). This connection is well established and forms the foundation of Hebbian learning theory. The authors fail to acknowledge or engage with this known property and instead conflate PCA-type Hebbian learning with the L2 norm stabilization.
- Hebbian learning, in particular nonlinear forms, implements unsupervised feature learning objectives such as PCA, ICA, or sparse coding (e.g. Oja 1982; Oja 1991; Clopath et al. 2010; Zylberberg et al. 2011). These rules learn statistical structure beyond norm stability. The paper’s framing ignores this and instead treats any gradient-norm correlation as Hebbian, which overlooks decades of theoretical and experimental work.
- The claim that anti-Hebbian alignment results from noise or that Hebbian signatures imply hidden global optimization lacks quantitative or mechanistic justification. Real synaptic plasticity involves nonlinear, spike- or voltage-dependent mechanisms (e.g., BCM, triplet STDP, Clopath rules) absent from this discussion.
- The paper ignores key models showing anti-Hebbian plasticity as a structured, biologically grounded mechanism for decorrelation and stability, not a noise artifact (Vogels et al. 2011; King et al. 2013).
- The manuscript overlooks nonlinear Hebbian learning frameworks that unify Oja’s, BCM, and triplet-STDP rules, as well as recent work linking these to ICA and sparse coding. These studies demonstrate that Hebbian-like rules do much more than stabilize norms, by extracting higher-order structure from data. The omission leads to incorrect generalization about Hebbian dynamics.

**Questions:**

1) Can the authors clarify whether their “Hebbian alignment” has any relationship to the PCA or ICA objectives known to be optimized by Hebbian-like rules? If not, how is it meaningful beyond norm equilibrium?
2) How does the proposed theory differentiate between trivial weight-norm alignment and genuine correlational structure learning? Would linear networks on whitened data still exhibit the claimed effects?

---

> ### Author Response · Authors · 2025-11-22
>
> We appreciate your deep familiarity with the Hebbian learning literature and the detailed critique. While you raise a number of very good points and interesting avenues for further research, we believe that you have had a subtle yet important misunderstanding of this paper’s contributions and scope (see Summary Rebuttal). We have replied to each of the points you raised and have worked to improve our manuscript with your feedback to avoid any ambiguity.
>
> We do not claim to present a complete functional account of Hebbian or anti-Hebbian learning in the brain, nor do we argue that L2 decay implements classical “Hebbian computation.” Our goal is narrower: to name a novel and unknown reason why one **could** Hebbianity during learning, caution neuroscientists about this possibility. This has been clear from our original statement of conclusion, which we quote here:
>
> C1: “*First, our results show that Hebbian and anti-Hebbian plasticity can emerge as regimes of gradient-based optimization, **in addition to** their conventional role as fundamental learning mechanisms.”*
>
> C2*:“Second, the presence of Hebbian or anti-Hebbian signatures in neuro-physiological data **need not be interpreted as evidence against** global error-driven optimization in the brain.*”
>
> The key message to neuroscientists is that the emergent and conventional Hebbian dynamics are difficult to distinguish and thus an important open problem.
>
> These points have also been clear from our discussions throughout the paper. In fact, as shown in Figure 4 (Figure 18 in the revision), explicit Hebbian update rules such as Oja’s do not align with SGD. This is why we used the terms Hebbian alignment and Hebbian dynamics and not computation, since it does not seem to be the case that SGD is actually doing an explicit conventional Hebbian update.
>
> Several of your critiques assume we aim to explain full biological plasticity or to subsume established Hebbian models, whereas our contribution is to show how such signatures arise generically across a broad class of learning rules and on learning tasks that are more complex than the unsupervised PCA/ICA objectives that modern Hebbian models tend to optimize. These findings are compatible with models that provide a mechanistic implementation for Hebbian learning in the brain. Rather, we are showing that learning processes and dynamics that are not functioning mechanistically like a Hebbian update can still look Hebbian.
>
> We aim to caution against assuming that data with a Hebbian appearance must arise from a mechanism that operates as a Hebbian rule, and additionally that the Hebbian component isn’t always the most important component in the optimization process.
>
> The revisions clarify this scope and the distinction between alignment phenomena and classical Hebbian objectives. We address these points in more detail below.
>
>
> **W1:** Norm stabilization versus Hebbian computation
> > The paper equates the norm-stabilizing effect of L2 regularization with Hebbian dynamics. L2 weight decay enforces contraction of weight norms, while Hebbian learning refers to directional correlation between pre- and post-synaptic activity (e.g., feature extraction). The alignment the authors observe is a trivial consequence of regularization equilibrium (($\nabla_\theta \ell + \gamma W = 0$)), not evidence of Hebbian computation.
>
>
> We do not equate or claim to equate them; this is a misunderstanding of our work, and we have made changes in the revision to clarify. Our original manuscript was clear about this point. See our conclusion C1, our paper is about proposing a new mechanism for Hebbian dynamics, “in addition to” the conventional ones. Namely, we claim there are two parallel mechanisms that could confound experimentalists (and thus deserve a lot more academic attention):
>
> 1. Conventional: functional Hebbian learning, fire together, wire together (Oja 1982; Oja 1991; Clopath et al. 2010; Zylberberg et al. 2011, …)
> 2. Emergent: Any regularized learning signal
>
> We never claimed either one is correct. Our claim is that both could happen in theory; they are challenging to distinguish experimentally, and biological experiments need to be careful when interpreting data.

---

> > ### Author Response · Authors · 2025-11-22
> >
> > **W2:** Oja, PCA, and the foundation of Hebbian theory
> > > Classical linear Hebbian rules, when combined with weight normalization or decay, are mathematically equivalent to stochastic gradient descent on the PCA objective (Oja, 1982). This connection is well established and forms the foundation of Hebbian learning theory. The authors fail to acknowledge or engage with this known property and instead conflate PCA-type Hebbian learning with the L2 norm stabilization.
> >
> > We do cite some of Oja’s work and fail to see how it in any way undermines our contributions. We are not optimizing for a PCA objective; in fact, we explore both supervised regression and classification of deep networks. This is a very different category of learning problem. Still, we agree it would be beneficial to more clearly position our work with respect to PCA/ICA and have made changes as described in the next response.
> >
> > Again, our claim is the need to distinguish the two types of Hebbianity, not on the correctness of either:
> >
> > 1. Conventional: functional Hebbian learning, fire together, wire together (including these equivalences to PCA learning in specific scenarios)
> > 2. Emergent: Any regularized learning signal
> >
> > We never claimed either one is correct, as is impossible for our paper, which is primarily theoretical and computational. The need to distinguish the two for biological experiments is exactly our conclusion and not our weakness.
> >
> > **W3, W6, Q1:** Discussion of nonlinear Hebbian frameworks, ICA, and sparse coding, unsupervised statistical structure learning.
> > > W3: Hebbian learning, in particular nonlinear forms, implements unsupervised feature learning objectives such as PCA, ICA, or sparse coding (e.g. Oja 1982; Oja 1991; Clopath et al. 2010; Zylberberg et al. 2011). These rules learn statistical structure beyond norm stability. The paper’s framing ignores this and instead treats any gradient-norm correlation as Hebbian, which overlooks decades of theoretical and experimental work. W6:  The manuscript overlooks nonlinear Hebbian learning frameworks that unify Oja’s, BCM, and triplet-STDP rules, as well as recent work linking these to ICA and sparse coding. These studies demonstrate that Hebbian-like rules do much more than stabilize norms, by extracting higher-order structure from data. The omission leads to incorrect generalization about Hebbian dynamics. Q1: Does your Hebbian alignment metric relate to the PCA/ICA objectives optimized by Hebbian rules, and if not, how is it nontrivial?
> >
> > Thank you for these references. To reiterate, our central claim is the **need to distinguish** the two types of Hebbianity, not on the correctness of either:
> >
> > 1. Conventional: functional Hebbian learning, fire together, wire together (including these equivalences to PCA learning in specific scenarios)
> > 2. Emergent: Any regularized learning signal
> >
> > We never claimed either one is correct, as that would be impossible for our paper, given that it is primarily theoretical and computational. The need to distinguish between the two for biological experiments is exactly our conclusion, not our weakness.
> >
> > Additionally, PCA feature learning is an entirely different learning problem than the one we analyze in this work. As Figure 4 (Figure 18 in the revision) illustrates, even classical Hebbian rules (including correlation-based updates and Oja’s rule) do not recover the same features as SGD in our setting. Moreover, the problems we study are substantially more complex than typical PCA/ICA-like objectives and are explicitly supervised. While the cited works are extremely important contributions to modern Hebbian learning theory, they are not directly applicable to the questions our manuscript addresses.
> >
> > However, despite addressing a different learning problem and not challenging our claims, they do deserve some discussion to situate our contribution more clearly. We have therefore added a paragraph in the introduction when reviewing background on Hebbian learning, referencing them.

---

> > > ### Author Response · Authors · 2025-11-22
> > >
> > > **W4 and W5:** Anti-Hebbian alignment, noise, and existing models
> > > > W4: The claim that anti-Hebbian alignment results from noise or that Hebbian signatures imply hidden global optimization lacks quantitative or mechanistic justification. Real synaptic plasticity involves nonlinear, spike- or voltage-dependent mechanisms (e.g., BCM, triplet STDP, Clopath rules) absent from this discussion. W5: The paper ignores key models showing anti-Hebbian plasticity as a structured, biologically grounded mechanism for decorrelation and stability, not a noise artifact (Vogels et al. 2011; King et al. 2013).
> > >
> > > We do not intend to argue that biological anti-Hebbian plasticity is “just noise.” Rather, we show that in a broad range of optimization settings, adding stochasticity to parameters or updates leads to a robust anti-Hebbian projection of the learning signal once noise dominates over decay.
> > >
> > > We will add explicit discussion of Vogels et al. and King et al., and clarify that our work is a complementary viewpoint that is not incompatible with other anti-Hebbian mechanisms.
> > >
> > > **Q2:** Trivial norm alignment, structure learning, and whitened inputs
> > > > How does the proposed theory differentiate between trivial weight-norm alignment and genuine correlational structure learning? Would linear networks on whitened data still exhibit the claimed effects?
> > >
> > > Our theory does not aim to diagnose whether the network is performing conventional or Hebbian learning, and our key claim and conclusion is exactly that future works need to find ways to distinguish them. In the revision, we updated the SRE description to clarify that the random inputs are i.i.d. and isotropic Gaussian (and thus whitened).

---

### Official Review · Reviewer_L1Cy · 2025-11-01

**Soundness:** 2
**Presentation:** 3
**Contribution:** 2
**Rating:** 4
**Confidence:** 3

**Summary:**

This work explores under what conditions Hebbian and anti-Hebbian learning signal emerges from gradient-based optimization rules. They thereotically and experimentally find that using L2 weight decay, a broad class of learning rules exhibit Hebbian-like alignment of the learning signal near stationarity. With sufficiently strong noise, the alignment flips to anti-Hebbian. However, experiments on more broad class of neural network architecture are missing.

**Strengths:**

- A mathematical framework  shows why any learning rule with weight decay should exhibit Hebbian-like alignment at stationarity
- In linear settings (and empirically in nonlinear ones) the work shows that sufficiently large parameter/gradient noise reverses alignment.

**Weaknesses:**

* The paper claims that Hebbian/anti-Hebbian signals may be auxiliary emergent phenomena of complex learning rules. However, this claim is too strong. The presented theory and experiments more naturally support a stronger statement: Hebbian learning signals can arise from gradient-based rules under L2 weight decay regularization. Or, Hebbian-like updates can be implemented by multiple learning paradigms.

* The current validation is limited to small networks. The paper should include larger-scale models (e.g., ResNet, compact LLMs) and analyze how depth, initialization, activation functions, or others shape Hebbian versus anti-Hebbian orientation. Demonstrating the effect across a broader model family would substantially strengthen the conclusions.

* The paper should investigate the trade-off between the L2 regularization coefficient and task performance, and quantify how “more Hebbian-like” alignment relates to accuracy/generalization.

**Questions:**

* The theoretical analysis focuses on L2 weight decay. Although L1 and Dropout are mentioned, the manuscript should make explicit what learning directions these regularizers induce and how they compare to L2 in driving Hebbian or anti-Hebbian tendencies.

---

> ### Author Response · Authors · 2025-11-22
>
> Thank you for your review of our manuscript. We believe you missed a few details in figures present in the appendix and the end of the text that address most of the concerns you raise. However, if you have suggestions on how to clarify these points to the reader so it is clearer on the first read we would appreciate your recommendations.
>
> **W1:**
> > The paper claims that Hebbian/anti-Hebbian signals may be auxiliary emergent phenomena of complex learning rules. However, this claim is too strong. The presented theory and experiments more naturally support a stronger statement: Hebbian learning signals can arise from gradient-based rules under L2 weight decay regularization. Or, Hebbian-like updates can be implemented by multiple learning paradigms.
>
>
> The alternative claims you propose (which we assume you meant as weaker, not "stronger") are accurate and supported by our work. However, the first claim is as well. Table 1 shows experiments across learning rules that are not solely gradient-based, yet still exhibit strong alignment. However, final performance on learned tasks can differ substantially, since Hebbian learning is unsupervised. This implies that despite high alignment, the difference between gradient signals carries crucial learning information. Thus, the Hebbian learning signal is emergent and can be very strong—even in random learning rules, provided weights approach stationarity—yet it is not the most important learning signal present, and thus may be auxiliary to the more important one.
>
> **W2:**
> > The current validation is limited to small networks. The paper should include larger-scale models (e.g., ResNet, compact LLMs) and analyze how depth, initialization, activation functions, or others shape Hebbian versus anti-Hebbian orientation. Demonstrating the effect across a broader model family would substantially strengthen the conclusions.
>
> We do show experimental results for most of the architectures you suggest in this weakness. See Table 1 for transformer results, and Figure 10 (Figure 9 in revision) for scaling results. While we didn’t originally run experiments on ResNet, we did run CNNs and explored the impact sparsity more generally on alignment in Figure 11 (Figure 10 in revision). Sparsity, CNNs, and scale all seem to increase Hebbian alignment. We have run additional experiments with ResNet and added that to the appendix (Figure 11 in the revision).
>
> **W3:**
> > The paper should investigate the trade-off between the L2 regularization coefficient and task performance, and quantify how “more Hebbian-like” alignment relates to accuracy/generalization.
>
> We do show an analysis of this question in Figure 6 (Figure 5 of the revision). It seems that validation performance is often best with low alignment. While establishing if this relationship is causal is outside the scope of this work, it would be an exciting avenue of future research.
>
> **Q1:**
> > The theoretical analysis focuses on L2 weight decay. Although L1 and Dropout are mentioned, the manuscript should make explicit what learning directions these regularizers induce and how they compare to L2 in driving Hebbian or anti-Hebbian tendencies.
>
> We do show the directions that these regularizers induce and how they compare to L2 in Figure 16, though a more comprehensive analysis of other regularization techniques is outside the scope of this work.

---

### Official Review · Reviewer_etsD · 2025-11-01

**Soundness:** 3
**Presentation:** 3
**Contribution:** 2
**Rating:** 4
**Confidence:** 3

**Summary:**

The paper proposes the idea that
 phenomenologically Hebbian and anti-Hebbian plasticity can emerge as byproduct of much more general learning rules (gradient based) that include weight decay (a contractive force) and/or noise (an expansive force).


The authors postulate that gradient descent training with weight decay (especially with high weight decay values) results into weight updates that are similar to those happening under purely Hebbian learning.
They offer a clear argument that since weight decay is contractive, the learned gradient part of the update must be expansive on average, and an expansive, rank-1-looking update will look Hebbian. That is an interesting explanation for why Hebbian-looking updates can show up during learning with weight decay even when the underlying algorithm is not Hebbian. They find that  stronger weight decay, larger learning rate, and larger batch size
lead to better alignment between gradient-based and Hebbian weight updates.
Moreover they find that strong noise in learning results in a learning signal that is anti-Hebbian. They mention that when noise co-exists with weight decay there is a competition between the two forces that contribute Hebbian and anti-Hebbian aligned updates, and they identify a “phase transition” when the interplay between these two forces changes polarity.

In general the authors try to push the argument that Hebbian and anti-Hebbian plasticity might be a byproduct or components of a more general gradient-like weight optimisation. While I find it an interesting argument, I find their evidence and argumentation not strong enough to support this argument, while their theory mostly holds for stationary states. This argumentation requires stronger biological anchoring and a clearer mapping from the employed weight decay to biological decay processes.

**Strengths:**

- The authors have performed an extensive number of numerical experiments where they explore the alignment of the weight updated with the Hebbian weight updates for different parameter values of weight decay, noise amplitude, network sparsity, network size, learning rate, batch size, and for different learning rules (namely Adam, stochastic gradient descent, direct feedback alignment, and randomNN) and regularisers.
- The authors explicitly demonstrate that is the gradient part of the weight update that aligns with the Hebbian updates and not the full weight update, and thus there is no issue with weights growing infinitely large (since there is also a weight decay)
- The authors propose an interesting theory: gradients under realistic constraints project onto Hebbian-looking directions, and thus observing Hebbian plasticity in experiments does not rule out gradient-like learning.

**Weaknesses:**

- I feel the assumption of stationarity  does the heavy lifting in the arguments of the paper, and there is no concrete proof or evidence whether this Hebbian alignment arguments hold also in the out-of equilibrium state.
- the title of the paper overstates the finding, since in essence this holds only under the assumption of stationarity and weak coupling.
- No statistical reporting in many of the figures, the results seem like single runs (or the authors omit to mention over how many realisations they average).

**Questions:**

# Questions


- In figure 15 you are trying to show that the alignment persists all over training, however isn’t the Hebbian alignment of the learning signal a bit too weak throughout this experiment?
- What do the blue and red lines in Figure 3 left indicate? Is it Layer 1 and Layer 2 weight update alignment with Hebbian? Please put a legend key.
- What experiments do you think one could perform to validate your proposal?  if Hebbian signatures can be a by-product of many learning rules, which empirical measurement/experiment would falsify/validate the argument?
- You show that any update rule with decay will, on average, look Hebbian. Does that mean that much of the experimental evidence for Hebbian STDP could be reinterpreted as "we only ever observed the projected, stationary part of a richer gradient estimator"?
- You identified a Hebbian - anti-Hebbian transition controlled by (noise, weight decay). Can this be re-read as having Hebbian-like updates when the gradient estimation is clean, and anti-Hebbian when the estimation is noisy, i.e. interpret the boundary as a quality-of-gradient axis?
- In [1]  the authors mention heterosynaptic pathways as ways to direct gradient information. In the paper you show that heterosynaptic  rules also become Hebbian under decay. Does that mean these anatomical pathways could be there mainly to improve the pre-projection gradient, with Hebbianity just the surface readout?
- In Figure 16 caption the authors mention that batch normalisation seems to have anti-Hebbian effect, however as I understand the plot the effect is very small (alignment value <-0.1). Can you comment on how you support this statement?



### Typos and other comments:

- Line 86: Eq. equation
- In eq. 4 what is $\ell$?
- Line 174: “with” is missing

### Papers to be cited

[1] Richards, Blake Aaron, and Konrad Paul Kording. "The study of plasticity has always been about gradients." The Journal of Physiology 601.15 (2023): 3141-3149.

---

> ### Author Response · Authors · 2025-11-22
>
> Thank you for your review of our paper, and we appreciate your detailed critique. See our Summary Rebuttal for a high-level explanation of our changes. We have addressed the points you raised in more detail below.
>
> **W1:** Stationarity assumption
>
> > I feel the assumption of stationarity does the heavy lifting in the arguments of the paper, and there is no concrete proof or evidence whether this Hebbian alignment arguments hold also in the out-of equilibrium state.
>
> We agree that our formal derivation focuses on stationary states. This was a deliberate choice, since it yields a simple and general condition that applies across architectures and learning rules. However, we do not intend to claim that Hebbian or anti-Hebbian alignment is only a stationary phenomenon.
>
> Empirically, we have already examined alignment throughout training for a range of models and tasks. For example, Figure 15 tracks the Hebbian alignment of the learning signal over the full course of optimization and shows that alignment emerges early, while the model is still learning, and persists rather than appearing only at convergence. We revised the text to emphasize that the stationary analysis provides intuition that is borne out across training in practice. Motivated by your comment and by td9e, we added an appendix derivation (C5) for out-of-stationarity.
>
> **W2:** Title strength
>
> > the title of the paper overstates the finding, since in essence this holds only under the assumption of stationarity and weak coupling.
>
> Thank you for bringing this up. We agree that the title might have been a little too strong. We now modify our title with the qualifiers “Homeostatic” and "Regularized" to emphasize the stationary aspect. As described in our response to W1, the claim of Hebbian dynamics often emerging prior to stationarity is supported by the manuscript; we do not ever claim Hebbian computation fully governs the optimization process. We aim to say that optimizing objectives with regularization can produce dynamics that resemble Hebbian updates as they approach stationarity, even when the underlying learning rules are not Hebbian. We believe the revised title will more accurately communicate that, though we are happy to hear further suggestions.
>
> **W3:** Statistical reporting
>
> > W3: “No statistical reporting in many of the figures, the results seem like single runs (or the authors omit to mention over how many realisations they average).”
>
> Alignment curves in the main text and appendix are averaged over 10 random seeds, and within each run, we average alignment across many batches. Because the trends are very robust, the resulting standard deviations are small, and the std bars are often hidden by the markers. We appreciate you pointing this out, since we did originally overlook running Figure 3 on multiple seeds and have since done so and updated the figure with std bars. We realize this is not clear from the current figure captions. In the revision, we have stated the number of seeds explicitly in each relevant caption and made special note of the markers obscuring some std bars where applicable.
>
> **Q1:** Strength of alignment in Figure 15
> > In figure 15 you are trying to show that the alignment persists all over training; however, isn’t the Hebbian alignment of the learning signal a bit too weak throughout this experiment?
>
> In this experiment, we use a relatively small weight decay. As we show elsewhere in the paper, Hebbian alignment increases with the strength of decay, so one should not expect alignment values close to 1 in this regime. The point of Figure 15 is not to show maximal alignment but to show that a consistent positive Hebbian alignment persists over training, in line with the theoretical prediction. We will clarify this in the caption.
>
>
> **Q2:** Legend and layer labels in Figure 3
>
> > What do the blue and red lines in Figure 3 left, indicate? Is it Layer 1 and Layer 2 weight update alignment with Hebbian? Please put a legend key.
>
> Yes, the colored lines correspond to different layers. In the current version, this is only stated in the caption. In the revision, we updated the figure to use Layer instead of L in the legend.

---

> > ### Author Response · Authors · 2025-11-22
> >
> > **Q3:** Validation and falsification experiments
> >
> > > What experiments do you think one could perform to validate your proposal? If Hebbian signatures can be a by-product of many learning rules, which empirical measurement/experiment would falsify/validate the argument?
> >
> > We have provided substantial empirical support for the theory presented in our work already, so we are interpreting this question to be a biological one: “What wet lab experiments could be run in the brain to falsify the theory we present in this work?”
> >
> > This is a very challenging question, and gets exactly to the cautionary point we are trying to raise; on the surface, both causes of Hebbianity can look very similar, and more scholarly work needs to be directed towards how to distinguish them experimentally. We believe a comprehensive experimental validation is a subject for future work; however, we do provide some discussion in the neurobiology paragraph of Section 3. We have added some content in the revision and quoted an excerpt from the original here:
> >
> > “The learning signal is a fast process; it is likely to, for example, come from other neurons and take the form of electric currents and spikes (Lillicrap et al., 2020). The weight decay, however, is a much slower biochemical process and directly corresponds to the changes in the biochemical properties of synapses (such as a spine shrinkage (Stein et al., 2015)). Therefore, the biological realizations of these two processes are likely to take different forms and can be separately measured. This makes it particularly important to have a theory for the learning signal component of the update, as this can be directly measured through LTD and LTP experiments of Hebbian plasticity (e.g., see (Zenke & Gerstner, 2017)).”
> >
> > As for a more detailed experiment, our analysis predicts a systematic Hebbian to anti-Hebbian transition as effective noise is increased relative to decay, with the boundary quadratic in noise strength (as in Figure 5). An experimental test would be to:
> >
> > - measure a Hebbian alignment metric for synaptic changes under controlled stimulation protocols,
> > - systematically manipulate effective synaptic noise and homeostatic decay processes, and
> > - test whether there is a reproducible transition from Hebbian to anti-Hebbian alignment whose location shows a roughly quadratic dependence on noise.
> >
> > Observing such a transition with the predicted qualitative and approximate quadratic dependence would support our optimization-based explanation in a biological context. Failing to see any systematic dependence of Hebbian versus anti-Hebbian signatures on noise and decay, or finding a boundary with a clearly different scaling, could argue against it.
> >
> > **Q4:** Interpretation of Hebbian STDP evidence
> >
> > > You show that any update rule with decay will, on average, look Hebbian. Does that mean that much of the experimental evidence for Hebbian STDP could be reinterpreted as ‘we only ever observed the projected, stationary part of a richer gradient estimator’?
> >
> > Essentially yes.
> >
> > But, to clarify, we do not claim that the brain literally implements a full gradient estimator, nor that existing evidence for Hebbian STDP is entirely an artifact of observing this projected component. However, our results are consistent with the view that, for any stable learning rule with an effective decay term, the component of the learning signal that survives at stationarity must have a Hebbian-like projection.
> >
> > We see our work as raising a cautionary point: strong Hebbian alignment of measured plasticity does not uniquely imply that the underlying learning rule is a pure Hebbian rule. It is compatible with richer learning dynamics whose stationary component has a Hebbian projection. Distinguishing between these possibilities requires experiments that go beyond stationary measurements and wet lab validation. We will make this interpretational nuance explicit in the discussion.
> >
> >
> > **Q5:** Hebbian to anti-Hebbian transition as a quality of the gradient axis
> >
> > > You identified a Hebbian-anti-Hebbian transition controlled by (noise, weight decay). Can this be re-read as having Hebbian-like updates when the gradient estimation is clean, and anti-Hebbian when the estimation is noisy, i.e. interpret the boundary as a quality-of-gradient axis?
> >
> > Yes, this is a reasonable way to interpret the transition within this optimization framework, though again, we don’t make the claim that this occurs in the brain for the reasons above. When the effective gradient estimate is relatively clean compared to the decay term, the learning signal aligns Hebbianly. As noise dominates, the alignment flips to anti-Hebbian. In that sense, the phase boundary can be viewed as a quality of the gradient axis in artificial networks.

---

> > > ### Author Response · Authors · 2025-11-22
> > >
> > > **Q6:** Heterosynaptic pathways and Hebbianity as surface readout
> > >
> > > > In [1] the authors mention heterosynaptic pathways as ways to direct gradient information. In the paper, you show that heterosynaptic rules also become Hebbian under decay. Does that mean these anatomical pathways could be there mainly to improve the pre-projection gradient, with Hebbianity just the surface readout?
> > >
> > > Essentially yes. While our results are compatible with this interpretation, that claim would require more evidence. In our analysis, heterosynaptic rules can produce learning signals that, after the effect of decay is taken into account, have a Hebbian-like projection. This suggests that anatomical pathways that route heterosynaptic information could primarily shape a more complex underlying learning signal, with Hebbian-like plasticity emerging as its stationary projection.
> > >
> > > Investigating this possibility in specific circuit models would require more detailed anatomical and biophysical assumptions than we consider here, but we have added this paper as a citation when discussing gradients in the brain.
> > >
> > > **Q7:** Batch normalization and anti-Hebbian effects
> > >
> > > > In Figure 16 caption the authors mention that batch normalisation seems to have anti-Hebbian effect; however, as I understand the plot, the effect is very small (alignment value <-0.1). Can you comment on how you support this statement?
> > >
> > > You are correct that the magnitude of the anti-Hebbian alignment with batch normalization is modest. Our intention was to note that the effect is consistently negative and reproducible across runs, not to claim a large quantitative impact. Figure 16 was included as a qualitative exploration of how other regularization techniques influence alignment for completeness, though it is a non-essential note. The main quantitative claims of the paper concern L2 weight decay and parameter noise.
> > >
> > > We will soften the wording in the caption and text to describe the batch normalization effect as small but consistently anti-Hebbian, and we will state explicitly that a more comprehensive analysis of other regularizers is outside the scope of the present work.
> > >
> > >
> > > **Typos and minor comments**
> > >
> > > - We will fix the “Eq. equation” typo on line 86 and define ($\ell$) explicitly in Eq. 4 as the loss function.
> > > - We will fix the missing “with” on line 174.
> > > - We will clarify the notation around Eq. 4 (adding parentheses and spaces), and expand the intermediate steps between Eqs. 4 and 5, and around Eq. 9, in an appendix.
> > >
> > > Thanks for pointing these out. We have addressed them in the revision.

---

### Author Response · Authors · 2025-11-21
**Summary Rebuttal**

Thank you to all the reviewers for your time and review of our manuscript. We are grateful for the feedback and suggestions that each of you have made, and have made a number of changes to the paper to improve its clarity. We outline some of the major points below and have responded in detail to each of your suggestions. We have also uploaded a revised version of the manuscript with all changes made to the text colored in blue.

A central misconception raised by multiple reviewers (**7Qh1**, **GwC8**, and **etsD**) is the misbelief that our paper claims to *explain* all forms of Hebbian and anti-Hebbian plasticity. Our actual claim is narrower: regularized optimization naturally produces Hebbian- and anti-Hebbian-like dynamics that can **coexist** with, rather than replace, conventional functional Hebbian mechanisms. This point has been clear from the conclusion of our original manuscript, in which we say: “Given that unsupervised adaptation and reinforcement are useful and widespread mechanisms, intrinsically Hebbian homosynaptic plasticity likely does exist in the brain. However, much of the existing experimental evidence for Hebbian and anti-Hebbian plasticity is often correlational and phenomenological (e.g., see Lamsa et al. (2007)), so it can be difficult to decide whether the underlying dynamics are actually Hebbian or are more complicated and only appear to be Hebbian.”

Reviewers **7Qh1** and **GwC8** interpreted our results as equating norm-stabilizing effects with Hebbian computation, whereas our contribution is to show how Hebbian *alignment* can arise from learning rules that are not themselves Hebbian. To address these points, we have made changes to the manuscript’s title, abstract, and introduction to clarify the scope. Several reviewers had other criticisms, which we also address with additional experiments, formal theoretical derivation, and discussions.

In summary, we have made the following additions and changes to the manuscript to address all concerns of reviewers:

1. Clarification of the scope of our work in the introduction and throughout the paper, as stated above (**7Qh1**, **GwC8**, **etsD**)
2. A formal formulation of the theory and an out-of-stationarity analysis in section Appendix C5 (**td9e, etsD**)
3. Analysis and proof for the generalized case where weight decay is different for different neurons in Appendix C4, which improves the biological plausibility of the theory (**7Qh1**)
4. Distinguishing from prior works in the introduction (**GwC8**)
5. Responding to **L1Cy**, we added ResNet experiments (Figure 11), though do note the original manuscript did already contain the other empirical suggestions that were made (Figures 10, 16, and Table 1)

Alongside other smaller revisions that clarify our notation, we believe that the reviews have significantly helped us improve the manuscript. The updated text underscores the key message, which is novel: Hebbian-like signatures in observed learning dynamics do not uniquely imply a Hebbian learning rule, and distinguishing conventional from emergent Hebbianity is an important, open, and experimentally crucial challenge.

---

### Meta-Review · Area_Chair_KhRq · 2026-01-04

**Summary:**

The paper is close to borderline and I propose a reject based on the following fact:
Reviewer 7Qh1’s concern about the lack of relationship between the roughly-Hebbian-aligned behavior and known neurophysiological experiments.

There are consequences of Hebbian learning in the way Hebbian theory predicts and are observed through neurophysiological experiments, such as PCA-like feature extraction. Do they appear in the studied learning mechanisms as well? Without such additional evidence, the claims remain weakly supported.

Some of the plots are based on single runs while it is claimed that the trend is present across different runs. There are ways to show it in the same plot and should have been done.

Notes for the authors: there seems to be a repeated misunderstanding among some of the reviewers regarding what the paper is about and how the reviewers perceived the paper. I suggest addressing this potential understanding in the paper from the get-go.

The word “Homeostatic”, used in the title, should also be mentioned in the introduction.

What is “std bars”? Std error or deviation?

**Reviewer Concerns:**

Reviewer L1Cy’s concern about using small networks is not fully addressed. Reviewer 7Qh1’s, who had a strong argument against the paper which I wrote below.

**Reviewer Scores:**

Reviewer etsD had two main concerns: the paper is mainly about the stationary case and that the results were perhaps for single runs. The authors clarified that the results are on multiple runs and they have results on the transient phase as well, where alignment still shows up, as shown in Section 5. Hence, the reviewer would have bumped up the score to accept.

Reviewer L1Cy had concerns about the statement being too strong, suggesting a weaker claim, that the evidence is shown on small networks, and that a connection to accuracy was not made (reviewer’s perception). The authors responded by noting that Transformers are used, accuracy connections are shown although no conclusive pattern was found,and the statement was slightly weakened by adding “regularized” in the title. Given that the variety of models and tasks are limited, it is unlikely that the scores would have increased.

Reviewer GwC8 brings substantial concerns about not incorporating existing work, theory, and results, e.g., on nonlinear Hebbian theory. This would require substantial effort to incorporate.

Reviewer 7Qh1’s main concern, as I understand, is whether uniform weight decay-like rule is really present in the brain. The most consequential criticism was this: “The fact that roughly-Hebbian-aligned changes would be necessary to counteract a hypothetical weight decay, in the long run, does not seem relevant to such controlled neurophysiology experiments?” The authors would need more work to address these.

Reviewer td9e already increased the score to an accept.

---

### Decision · Program_Chairs · 2026-01-26

Reject